**Fungal Genetics and Genomics**

# Genetic and environmental determinants of multicellular-like phenotypes in fission yeast

Bence Kövér (ID) ,[1] Céleste E. Cohen,[1] Ladislav Seres,[1] Saniya Raut,[1] Markus Ralser (ID) ,[2,3,4] Benjamin M. Heineike (ID) ,[1,2,]*
Jürg Bähler (ID) [1,]*

[1]Department of Genetics, Evolution and Environment, Institute of Healthy Ageing, University College London, London WC1E 6BT, United Kingdom
[2]Nuffield Department of Medicine, The Centre for Human Genetics, University of Oxford, Oxford OX3 7BN, United Kingdom
[3]Department of Biochemistry, Charité Universitätsmedizin Berlin, 10117 Berlin, Germany
[4]Max Planck Institute for Molecular Genetics, 14195 Berlin, Germany

*Corresponding authors: Benjamin M. Heineike, Department of Genetics, Evolution and Environment, Institute of Healthy Ageing, University College London, London WC1E 6BT, United Kingdom. Email: b.heineike@ucl.ac.uk; Jürg Bähler, Department of Genetics, Evolution and Environment, Institute of Healthy Ageing, University College London, London WC1E 6BT, United Kingdom. Email: j.bahler@ucl.ac.uk

Multicellular fungi have repeatedly given rise to primarily unicellular yeast species. Some of these, including the fission yeast *Schizosaccharomyces pombe*, can revert to multicellular-like phenotypes (MLPs). As MLP formation remains understudied in fission yeast compared with budding yeast, we aimed to narrow this gap. We developed high-throughput assays for two MLPs: flocculation and surface adhesion, which correlated in minimal media, suggesting a common mechanism. Using a library of 57 natural *S. pombe* isolates, we found that MLP formation varied widely across different nutrient and drug conditions. In a segregant *S. pombe* library generated by crossing an adhesive natural isolate with the standard laboratory strain, MLP formation correlated with expression levels of the transcription factor gene *mbx2* and several flocculin genes. Quantitative trait locus (QTL) mapping of MLP formation located a frameshift mutation in the *srb11* gene encoding cyclin C, a part of the Cdk8 kinase module (CKM) of the Mediator complex. Deletion of either *srb11* or *srb10* (encoding the Cdk8 kinase) resulted in MLP formation through upregulation of *mbx2*. Screening a library of 3,721 gene-deletion strains uncovered 31 additional genes involved in surface adhesion, including 15 genes not previously associated with MLPs in fission or budding yeast. Notably, deletion of *srb11*, unlike deletions of the 31 hits, did not compromise cell growth, which might explain its natural occurrence as a QTL for MLP formation. Our findings provide a comprehensive genetic survey of MLP formation in fission yeast and a functional description of a causal variant that drives MLP formation in nature.

**Keywords:** linkage study; quantitative genetics; *S. pombe*; evolution; adhesion; Mediator complex; cyclin C; transcription factor; flocculin

## Introduction

Yeast species are unicellular fungi in the phylum Ascomycota. Within the Ascomycota, fission yeasts are found in the Taphromycotina sub-phylum, while budding yeasts belong to the Saccharomycotina (Spatafora et al. 2017). Phylogenetic data suggest that the last common ancestor of these branches, which existed around 330 to 420 MYA (Sipiczki 2000), was multicellular but already capable of switching to planktonic growth (Nagy et al. 2014). The primarily unicellular lifestyle of yeasts then repeatedly evolved by deploying a conserved set of transcription factors in each clade (Nagy et al. 2014). Yet hundreds of millions of years later, yeasts still exhibit a range of multicellular-like phenotypes (MLPs) (Amoah-Buahin et al. 2005; Bähler 2005; Verstrepen and Klis 2006; Dodgson et al. 2009; Soares 2011; Cullen and Sprague 2012; Chow et al. 2019a). Two widely used model organisms, the budding yeast *Saccharomyces cerevisiae* and the fission yeast *Schizosaccharomyces pombe,* exhibit flocculation (formation of multicellular aggregates) in liquid media and filamentous growth on agar plates (Amoah-Buahin et al. 2005; Bähler 2005; Verstrepen and Klis 2006; Dodgson et al. 2009; Soares 2011; Cullen and Sprague 2012; Chow et al. 2019a). The latter is often

coupled with the ability to invade agar (Amoah-Buahin et al. 2005; Bähler 2005; Verstrepen and Klis 2006; Dodgson et al. 2009; Cullen and Sprague 2012; Chow et al. 2019a). The fission yeast *Schizosaccharomyces japonicus*, closely related to *S. pombe*, also forms long filaments (Sipiczki et al. 1998; Gómez-Gil et al. 2019; Papp et al. 2021). Research on flocculation in *S. cerevisiae* has been driven by its role in brewing, where the formation of flocs allows simple removal of biomass from each batch (Soares 2011). Besides *S. cerevisiae*, *Candida albicans* is the most studied yeast for MLPs such as flocculation (Bauer and Wendland 2007), filamentous growth, and the clinically important phenotype of biofilm formation (Douglas 2003; Gulati and Nobile 2016).

Importantly, these phenotypes are not *bona fide* multicellularity, as they are temporary, lack committed cell fates, and the constituent cells replicate independently rather than on a colony level (no germline). Nevertheless, filaments or clonal clumps (Koschwanez et al. 2013) are made of genetically related cells, and there is some evidence that aggregative flocs preferentially contain clonal cells, a phenomenon known as positive assortment (Smukalla et al. 2008). Experimental evolution towards aggregative multicellularity in *S. cerevisiae* has indeed identified the

repeated emergence of positive assortment (Stoy et al. 2025). As such, these structures might constitute a level at which natural selection can act (Libby and Rainey 2013). Here we use the term MLPs to refer to flocculation, surface adhesion, filamentous growth, invasive growth and biofilm formation, or any combination of these, in yeast species.

MLPs can give rise to emergent properties. Filamentation may facilitate foraging in nutrient-poor conditions (Amoah-Buahin et al. 2005; Bähler 2005; Verstrepen and Klis 2006; Soares 2011; Cullen and Sprague 2012). Alternatively, cell aggregates could share metabolic products of excreted enzymes as "public goods", thereby increasing local nutrient concentrations in otherwise nutrient-poor environments (Koschwanez et al. 2011, 2013). Indeed, at low sucrose concentrations, S. cerevisiae cells that clump together grow more efficiently than dispersed cells, primarily because they can share the products of the external sucrose invertase Suc2p, glucose and fructose (Koschwanez et al. 2011). Moreover, inner cells of biofilms and flocs can be protected from environmental insults by the outer cell layers. Data from S. cerevisiae support a role of flocs in protection against high concentrations of ethanol, hydrogen peroxide, antifungals and UV (Smukalla et al. 2008). Moreover, surface adhesion and aggregation might protect colonies from being consumed by macroscopic predators, as shown for S. cerevisiae consumption by *Caenorhabditis elegans* (Chow et al. 2019b).

Within a species, the regulators and effectors for different MLPs often overlap. The transcription factor Flo8p controls flocculation, filamentous growth and invasive growth in S. cerevisiae (Liu et al. 1996), and Mbx2 plays a similar role in S. pombe, although the two transcription factors are not orthologs (Matsuzawa et al. 2012). The cell-surface adhesion protein Flo11p is required for both invasive and filamentous growth, and, to some extent, for flocculation in S. cerevisiae (Lo and Dranginis 1998). Similarly, the dominant flocculin Gsf2 is required for invasion, filamentous growth and flocculation in S. pombe (Matsuzawa et al. 2011; Kwon et al. 2012). These observations, and other examples (Bauer and Wendland 2007; Převorovský et al. 2009a; Chow et al. 2019a, 2019b), suggest deep evolutionary and mechanistic connections between various MLPs and justify studying them together.

In S. pombe, sexual flocculation occurs between cells of opposite mating types as the first step of mating (Miyata et al. 1997). Nonsexual flocculation occurs outside of mating, between clonal cells regardless of mating type (Miyata et al. 1997) and depends on cell-surface flocculins that bind cell wall galactose residues in a $Ca^{2+}$-dependent manner (Tanaka et al. 1999; Matsuzawa et al. 2011). S. pombe is also capable of forming filaments, which can invade solid media (Amoah-Buahin et al. 2005; Dodgson et al. 2009; Převorovský et al. 2009b; Matsuzawa et al. 2012). To quantify filamentation and invasion in S. pombe, assays have been developed to quantify the ability of cells grown on agar plates to resist washing (Amoah-Buahin et al. 2005; Dodgson et al. 2009; Převorovský et al. 2009b; Matsuzawa et al. 2011, 2012; Kwon et al. 2012; Cullen 2015; Su et al. 2018).

During nonsexual flocculation, cell adhesion in S. pombe is primarily mediated by the flocculins Gsf2 and Pfl2-9 (Matsuzawa et al. 2011; Kwon et al. 2012). These flocculins are positively regulated by the transcription factors Mbx2 and Cbf12, and are repressed by Gsf1 and Cbf11 (Tanaka et al. 1999; Převorovský et al. 2009a; Kwon et al. 2012; Matsuzawa et al. 2012 , 2013; Marešová et al. 2024). Another important aspect of MLP formation is the control of cell separation after mitosis. The genes coding for enzymes participating in septum digestion are activated by the transcription factor Ace2 (Alonso-Nuñez et al. 2005). The *ace2* gene and the Ace2 targets are positively and negatively regulated by the

transcription factors Sep1 and Fkh2, respectively (Rodríguez-López and Bähler 2015; Suárez et al. 2015), and this pathway could contribute to filamentation (Bähler 2005). Nitrogen starvation can trigger filament formation, but this requires a carbon source such as glucose, which activates the cAMP/PKA pathway (Amoah-Buahin et al. 2005). Asp1, a kinase producing the inositol-pyrophosphate IP8, is required for filamentation through the cAMP pathway, and overproduction of IP8 leads to flocculation (Pöhlmann and Fleig 2010). Interestingly, IP8 signaling is also associated with *mbx2* upregulation (Sanchez et al. 2019). High iron concentration also triggers surface adhesion and invasion of growth media (Převorovský et al. 2009b). Furthermore, deletions of members of the Mediator Cdk8 kinase module (CKM) and of some ribosomal genes (Li et al. 2013; Liu et al. 2015) cause flocculation, while deletions of genes involved in mitochondrial gene expression (Wang et al. 2017; Su et al. 2018) cause both flocculation and filamentous growth.

Much of the research on MLPs has been developed in the *Saccharomycotina* clade with studies in the budding yeasts S. cerevisiae and C. albicans (Verstrepen and Klis 2006; Soares 2011; Brückner and Mösch 2012; Cullen and Sprague 2012; Gulati and Nobile 2016), while much less is known about MLPs in S. pombe (Dodgson et al. 2009). A deeper understanding of the mechanisms behind MLP formation in S. pombe could provide insights into the latent capacity of yeast species to revert to ancestral multicellular phenotypes. To this end, we first analyze existing data from model organism databases and show that while S. pombe shares several regulatory proteins for MLP formation with S. cerevisiae and C. albicans, downstream effector cell-adhesion proteins are mostly not conserved between the three species. These results suggest novel mechanisms for MLP formation in S. pombe. In our lab, we observed that the natural isolate JB759 flocculates and weakly adheres to glass flasks in minimal media. Here we screen for MLP formation across 57 nonclonal natural isolates (Jeffares et al. 2015), and find it to vary widely between strains and nutrient conditions. To understand the genetic basis of MLP formation in JB759, we apply a quantitative genetic approach, revealing that MLP formation correlates with the expression of *mbx2* and flocculin genes. Through quantitative trait locus (QTL) mapping, we identify a causal frameshift mutation in *srb11*, the cyclin in the CKM of the mediator complex. Deletion of *srb11* and its associated kinase gene, *srb10*, caused an increase in *mbx2* expression, and MLP formation in the *srb11* deletion strain depended on *mbx2*. To identify additional factors involved in MLP formation, we screened a genome-wide deletion library (Kim et al. 2010; Malecki and Bähler 2016) and a library of long noncoding RNA deletions (Rodriguez-Lopez et al. 2022). We validated 3 known and uncovered 28 new genes involved in surface adhesion on minimal media. Of these, 13 had no previous annotation to MLPs in S. cerevisiae or C. albicans, while being genetically conserved. Interestingly, none of the adhesive natural isolates possess a null-mutation in the genes we identified as hits in the screen. We conclude that a null-mutation in *srb11* maintains better growth efficiency compared with the hits revealed in the screen, likely explaining why only the former has been observed as a natural QTL for MLP formation.

## Materials and methods
### Yeast strains
#### Segregants

Clément-Ziza et al. (2014) created a segregant library by mating Leupold's lab strain 968 $h^{90}$ (or JB50) and the South African natural

**Table 1.** Strains generated in this study.

| Engineered strain name | Background | Genotype |
|---|---|---|
| srb11 CRISPR knock-out strain (JB1785, JB1786) | JB50 (968 h90) | srb11Δ (CRISPR) |
| srb11 and mbx2 double-knock-out strain (JB1782, JB1783, JB1784) | srb11Δ::Kan from deletion library | srb11Δ::Kan mbx2Δ (CRISPR) |
| mbx2 overexpression strain (JB2102) | JB22 (972 h−) | prRPL2102-mbx2-tTEF, his3, NatMX::his3 |
| mbx2 overexpression strain, (JB2103) | JB50 | prRPL2102-mbx2-tTEF, his3, NatMX::his3 |
| sfGFP overexpression strain, (JB2104) | JB22 | prRPL2102-sfGFP-tTEF, his3, NatMX::his3 |
| sfGFP overexpression strain, (JB2105) | JB50 | prRPL2102-sfGFP-tTEF, his3, NatMX::his3 |
| med10Δ (JB2106) | JB22 | med10Δ::NatMX |
| med12Δ (JB2107) | JB22 | med12Δ::NatMX |
| med13Δ (JB2108) | JB22 | med13Δ::NatMX |
| med19Δ (JB2109) | JB22 | med19Δ::NatMX |
| med27Δ (JB1996) | JB22 | med27Δ::NatMX |
| srb10Δ (JB2111) | JB22 | srb10Δ::NatMX |

isolate Y0036 (or JB759). Together with the two parental strains, we assayed 54 segregants from this segregant library, out of which two strains were identified as identical to other strains in the library upon sequencing. Segregants were named Rxx, e.g. R45.

## Natural isolates

Jeffares et al. (2015) collected 57 nonclonal natural isolates which span the natural diversity of S. pombe. Natural isolates were named JBxxxx, e.g. JB1207.

## ncRNA deletion library

Rodriguez-Lopez et al. (2022) created a library of null-mutants for 141 different long noncoding RNA (lncRNA), each with multiple biological and technical replicates. During the confirmation step of the deletion library screen, all replicates of SPNCRNA.1234Δ and SPNCRNA.900Δ were verified by PCR.

## Genome-wide prototrophic deletion library

The Bioneer V5 auxotrophic deletion library (Kim et al. 2010) was backcrossed with a wild-type strain to produce a prototrophic deletion library as detailed in Malecki and Bähler (2016). We used a copy of this library from Maria Rodriguez-Lopez, with multiple replicates for certain strains. All Mediator gene deletions were verified using PCR and gel electrophoresis. During the confirmation step of the deletion library screen, we chose a random set of 16 genes, out of which 14 were successfully verified with PCR.

## mbx2 overexpression strain

Overexpression strains were generated using a MoClo Golden Gate framework using parts from the Yeast Tool Kit (YTK) (Lee et al. 2015) (Addgene plasmid kit 1000000061) and the POMBOX extension (Hebra et al. 2024) (Addgene plasmid kit 1000000251) (Supplementary Table 14). We first constructed a GFP dropout vector plasmid (pLS007) that was designed to insert in the S. pombe his5 locus without damaging the his5 gene itself and carrying the NatMX marker. This plasmid included a part8b plasmid (pLS006) containing the his5 and its 5′ region from plasmid pAV0663 (Vještica et al. 2020), Addgene 133487). This PCR product was inserted into the YTK001 parts entry vector with a BsmBI Golden Gate reaction. pLS007 was then constructed using a Golden GateBsaI reaction using YTK002 (ConLS, type 1), YTK047 (GFP dropout, type 234r), YTK067 (ConR1, type 5), YTK078 (NatMX, type 6), POM033 (3′ homology region for S. pombe his5, type 8), YTK089 (ColE1 bacterial replication origin with Amp marker, type 8a) and pLS006 (his5 gene and 5′ homology region, type 8b). We then domesticated the mbx2 gene as a type 3 part (pHB001) using a BsMBI reaction including YTK001 and three PCR products

from the JB22 genome which amplified mbx2 in three sections to remove a Bsa1 cut site and a BsmBI cut site in the genomic sequence. This required changing the Leu23 CTC codon to CTA, and the Thr238 codon from ACG to ACC. As a control we also domesticated sfGFP as a type 3 part by amplifying the fluorophore from AV0327 (Addgene 133488) and inserting into YTK001. We then constructed plasmids to drive expression of either mbx2 (pHB002) or sfGFP as a control (pLS012) under the rpl2102 promoter using Bsa1 reactions including LS007, pPOM019 (rpl2102 promoter, type 2), pHB001 (mbx2, type 3) or pLS004 (sfGFP, type 3), and YTK052 (S. cerevisiae SSA1 terminator, type 4). We transformed these plasmids into JB22 (h−) and JB50 (h90) using a standard Lithium Acetate protocol (Murray et al. 2016). The resulting strains are shown in Table 1. For the transformation, we linearized 1 µg of plasmid using NotI and incubating at 37 °C for 1 h and heat inactivating for 20 min at 65 °C. After recovery in YES, transformants were selected on plates containing 100 µg/mL noursethricin. Colonies were screened for flocculation and fluorescence. We were unable to obtain transformants in which mbx2 was driven by the very high expression adh1 and eno1 promoters.

## Mediator subunit knock-out strains

Clean Mediator subunit knock-out strains (Table 1) were made by amplifying the NatMX marker from pFA6A-Nat-mx6 (Bähler et al. 1998) with 80bp homology arms using PCR and transforming JB22 (h−) strains with a standard Lithium Acetate protocol (Murray et al. 2016). Approximately 1 µg of PCR product was used for each transformation, and strains were selected on 100 µg/mL noursethricin after recovery in YES. Strains were verified by PCR to confirm integration of the NatMX marker at the correct location and deletion of the gene.

## CRISPR-edited strains

We created a deletion of srb11 in the JB50 background, and a deletion of mbx2 in an srb11Δ::Kan background taken from the genome-wide prototrophic deletion library (Table 1). Seamless CRISPR-Cas9 gene-editing was done using a published protocol (Rodríguez-López et al. 2016). Briefly, single-guide RNAs were inserted in the pMZ379 (Addgene 74215) plasmid using a PCR-based method, while homology templates were generated as large primer dimers also using PCR. To design the single-guide RNA and homology template, we used the CRISPR4P tool (http://bahlerweb.cs.ucl.ac.uk/cgi-bin/crispr4p/webapp.py) (Rodríguez-López et al. 2016). Furthermore, we checked the sgRNAs in Benchling ["Benchling (Biology Software)," no date] and chose the sequences with the most favorable on-target and

**Table 2.** Media.

| Name | Ingredients (for solid media add 2% agar) |
| --- | --- |
| Yeast extract media with supplements (YES—Rich media) | Yeast extract + adenine, uracil, histidine, lysine, leucine + 30 g glucose/l |
| Edinburgh minimal media (EMM) | EMM-N (Formedium) + 5 g NH4Cl/l |
| Phosphate starvation | EMM-P (Formedium) + 1.81 g NaCl/l |
| Nitrogen starvation | EMM-N (Formedium) + 0.05 g NH4Cl/l |
| EMM + ade (for segregants) | EMM-N (Formedium) + 5 g NH4Cl/l + 0.1 g ade/l |
| LB media for growing *E. coli* | LB (Formedium) |

off-target scores. The homology templates contained homologous regions at the edges of the gene of interest, allowing for knockouts.

Plasmids used for strain construction are listed in Supplementary Table 14.

## Media compositions and growth conditions

Media are listed in Table 2. Compounds were added at the following concentrations to these media: caffeine-10 mM, rapamycin-100 ng/mL. These concentrations were adopted from Rallis et al. (2013). RoToR HDA (Singer Instruments) compatible plates were poured using 40 mL of media. Strains were always grown at 32 °C, shaking at 160 rpm (liquid cultures in tubes or flasks) or 80 rpm (96-well liquid cultures) in an Infors HT Incutron incubator.

## Orthology analysis

Candidate genes involved in MLP formation were obtained from relevant Gene Ontology (GO) terms (Supplementary Table 1) from https://current.geneontology.org/products/pages/downloads.html. Orthology relationships between genes in *S. pombe* and *S. cerevisiae* were obtained from PomBase (Harris et al. 2022), while orthology relationships between genes from *C. albicans* and *S. pombe*, and *C. albicans* and *S. cerevisiae* were obtained from the *Candida* Genome Database (CGD) (Skrzypek et al. 2017). Such orthology annotations, in the case of PomBase, result from multiple algorithms and manual curation (Wood et al. 2012; Harris et al. 2022). In the analysis, genes were grouped into orthogroups to avoid confusion when accounting for paralogous genes (e.g. if one species had 5 versions of a gene, and another had 2, it was still counted in a single orthogroup). To extend these results, we also repeated this analysis after including genes from the Fission Yeast Phenotype Ontology (FYPO) (Harris et al. 2013), and the phenotype database from *S. cerevisiae* Genome Database (SGD; Cherry et al. 2012) and CGD. Since SGD and CGD do not have phenotype ontology, we obtained the respective tables of phenotypes, which were then filtered using keywords (Supplementary Table 1). We observed that *S. pombe* cell-adhesion proteins are not annotated as orthologs to other proteins in *S. cerevisiae* and *C. albicans*. To see whether the budding yeast cell-adhesion genes also lack orthologs in *S. pombe*, we examined cell-adhesion genes from *S. cerevisiae* and *C. albicans,* namely the *FLO* (FLOcculation) and *ALS* (Agglutinin Like Sequence) genes respectively.

## Sequence- and structure-based queries for protein conservation

We categorized proteins as either unique to *S. pombe* (including cell-adhesion proteins) or conserved across *S. pombe*, *S. cerevisiae* and *C. albicans*, based on annotations from model organism databases. We set out to independently verify this analysis using quantitative metrics. As a measure of protein conservation between *S. pombe* and *S. cerevisiae* or *C. albicans*, we used BLAST-P (Altschul et al. 1997) which compares protein sequences and Foldseek (van Kempen et al. 2024) which compares protein structures. First, protein sequences were fetched from Uniprot using the Uniprot IDs obtained from PomBase. BLAST-P queries were then submitted through the Python function NCBIWWW.qblast(), with the arguments: database = "nr", expect = 1000, entrez_query="xid237561[ORGN] OR txid5476[ORGN] OR txid559292[ORGN] OR txid4932[ORGN]" and hitlist_size = 1000. Alphafold-predicted protein structures were fetched from https://alphafold.ebi.ac.uk/. Foldseek queries were then submitted through the Foldseek API using the recommended command: *curl -X POST -F q=@PATH_TO_FILE -F 'mode = 3diaa' -F 'database[] = afdb-swissprot'* https://search.foldseek.com/api/ticket.

Both methods returned a list of candidate orthologs ranked by alignment scores. For all 25 "unique" proteins (including 14 cell-surface adhesion proteins (Fig. 1a), and a random set of 50 "conserved" proteins we queried putative orthologs with the highest BLAST-P and Foldseek scores (Supplementary Table 2). Statistical significance of the difference in these scores between "unique" and "conserved" proteins was assessed using a Mann–Whitney *U* test, separately for *S. pombe*–*C. albicans* and *S. pombe*–*S. cerevisiae* comparisons.

## High-throughput assays and *yeastmlp*

We developed two 96-well format high-throughput assays and a complementary software package for quantifying cell adhesion and flocculation. Both assays take advantage of the RoToR HDA colony-pinning robot (Singer Instruments), which can pin out yeast on agar plates in a 96-well arrangement. After conducting each assay, the data was analyzed using the "Yeast Multicellular-like Phenotype" analysis package *yeastmlp* (https://github.com/BKover99/yeastmlp). Before each assay, yeast were transferred to YES plates from −80 °C glycerol stocks and were grown for 3 d at 32 °C.

Our adhesion assay is a high-throughput variant of the conventional washing assay widely used in yeast literature (Cullen 2015). After pinning onto YES solid media from freezer stocks and incubating for 3 d at 32 °C, yeast were temporarily suspended in a 96-well plate filled with 200 µL EMM in each well. At this step, for the *mbx2* overexpression experiment in YES, 10 mM EDTA was added to disperse flocs. Also for the *mbx2* overexpression strains in YES, pinning from the RoToR did not transfer a sufficient amount of cells from the *mbx2* overexpression samples, as the colony did not adhere to the pinning pad. These colonies were transferred manually with a pipette tip. For all experiments, yeast were

then pinned to agar plates with desired media conditions using the "7 × 7 squares" program on the RoToR, which pins 49 yeast colonies in a square arrangement for all 96 strains on an agar plate. This arrangement was chosen because it prevented yeast colonies from being washed off as a single self-adhesive colony, and allowed proper adhesion to agar. Yeast cells were then grown for 4 d following which they were imaged on a flatbed scanner (Epson V800 Photo), washed with water (constant 35 mL/s flow rate, 1 s for each 7 × 7 square), and imaged again.

During analysis, *yeastmlp* takes a folder of pre and postwash images, a 96-well map of strains, and an example "filled-out" plate as arguments and returns adhesion values for each strain. To accurately discriminate between each square of cells, our algorithm creates a 96-well raw layout based on the example "filled-out" plate, where each square contains growing strains. This raw layout is then fitted to each image separately in the prewash folder. Because the layouts are freshly generated for each analyzed image, this method should be robust to images acquired using different scanners. Furthermore, the individual fitting of layouts to each image allows robust quantification even if images in the same folder are slightly misaligned (e.g. images from different quadrants of a flatbed scanner). Our assumption was, however, that pairs of pre and postwash images are not misaligned with respect to each other. Finally, the algorithm expects at least one negative control (empty square) on each plate to correct for background intensity.

Since cellular density allows less light to pass through, high cell density is represented by low pixel intensities. Therefore, our measure of cellular density was inverse pixel intensity. After segmentation with the fitted-layout, mean cell densities in each square were calculated. In rare cases, the scanner returned slightly higher density values for washed colonies than prewashing colonies; therefore, we decided to rescale all values to between 0 and 1 by dividing with the maximum inverse pixel intensity on a given image (meaning the darkest pixel). This resulted in robust and comparable estimates of colony density before and after wash in MLP-forming strains. We used a negative control (empty square) as a measure of background, which we subtracted from each measurement. Following this, the ratio of cell densities after and before washing allowed us to determine the fraction of cells sticking to the agar plate. Importantly, cells grown at the edges of the plate were more adhesive and produced less reliable measurements; therefore, the strains of interest were moved to the middle 60 positions. When a strain exhibited a prewash normalized pixel intensity value of < 0.1, it was considered not growing on the given plate, and was removed from downstream analysis. Strains that had a mean prewash normalized pixel intensity value of <0.1 were removed from downstream analysis entirely. Throughout the paper, example raw data of plates before, and after washing are shown using the viridis colormap, which appears perceptually uniform to the human eye (Nuñez et al. 2018).

For the flocculation assay, we used the "Archive" program on the RoToR to seed yeast cells in a non-TC treated 96-well plate in 200 µL liquid media in each well. Cells were grown for 2 d and were imaged on a Tecan Infinite M200 plate reader which allowed measurement of optical density in each well of the 96-well plate at up to 225 different locations.

During analysis, *yeastmlp* takes as arguments a folder of .csv files returned by the plate-reader, the square root of measurements per well (e.g. 15 for 15 × 15 measurements—used for file parsing purposes), a 96-well map of strains, and the location of the negative control well. The algorithm first subtracts the mean OD600 of the negative control from each well as a control for

background light absorption. Our measurement for flocculation was the coefficient of variation (CV or standard deviation/mean) of normalized optical density measurements in each well.

To validate our high-throughput flocculation assay, we also measured flocculation using a simpler filtering assay. For this, yeast cells were grown overnight in YES, diluted to $OD_{600} = 0.1$ and inoculated into tubes with 5 mL EMM. After 48 h of growth, cells were resuspended by gently flicking the tubes and the culture was poured through a 30 µm filter into a 50 mL tube "A". Flocculating cells stuck in the filter were washed into a second tube "B" and were completely resuspended using 10 mM EDTA and vortexing. After measuring the $OD_{600}$ of the content of both tubes, the ratio of flocculating to non-flocculating cells was determined as (OD(B)/(OD(A) + OD(B)). The relationship between the two assays, while nonlinear, was generally monotonic with a Spearman correlation coefficient of 0.24 ($P = 4E$ $-2$, Supplementary Fig. 1). Generally, we consider the filtering assay more accurate, while the plate-reader assay allows for higher throughput.

## Microscopy

JB50 and JB759 (Fig. 2a): Cells were grown from single colonies on YES plates over two days at 32 °C from single colonies in EMM or YES. After resuspending by shaking, 20 µL of cells were placed on a glass slide and covered with a coverslip. Cells were then imaged on a Zeiss Axio-Imager Z2 with a Hammamatsu OrcaFlash 4.0 Camera with ZenPro2.3 software under bright field illumination using 20× air, 40× oil and 100× oil objectives.

Natural isolates (Supplementary Fig. 4): JB50, JB759, JB914, and JB953 strains were inoculated in 5 mL YES and grown overnight. They were then diluted to OD 0.1 in 25 mL flasks in 10 mL of either YES, EMM, or Phosphate Starvation (EMM-P) media. Cells were then grown for 5 h at 32 °C. Then, 400 µL of cells were placed in a 96-well µ-plate with #1.5 polymer coverslip (Ibidi 89626) after the wells were washed with Milli-Q water. Cells were imaged in a Zeiss Cell Discoverer 7 microscope with an Axiocam 506 camera. For EMM and YES, a Plan-Apochromat 5×/0.35 objective with Castor Tubelens 2× optical zoom was used, and for phosphate starvation media, a Plan-Apochromat 20×/0.95 objective with no optical zoom was used. Cells were imaged using the TL Brightfield configuration settings with 10 ms exposure, at 10% intensity. For this experiment, photos of the flasks were also taken after 7 h and the following day. Images were resized, rescaled and cropped using custom Python scripts.

MBX2 overexpression (Supplementary Fig. 7): A single colony from each strain was grown overnight in 5 mL YES. The cells were then diluted into either YES or EMM to an OD of approximately 0.1 in a total volume of 1 mL in a deep well plate, shaking at 32 °C overnight. Cells were then diluted 40-fold into EMM or YES and incubated for 90 min at 32 °C. Then, 100 µL of cells were transferred into wells in a 96-well µ-plate with #1.5 polymer coverslip (Ibidi 89626) after the wells were washed with Milli-Q water. Cells were imaged in a Zeiss Cell Discoverer 7 microscope (UCL Center for Cell & Molecular Dynamics) with an Axiocam 506 camera using a Plan-Apochromat 20×/0.95 objective. Transmitted light was measured with a 10 ms exposure, with 2% intensity. GFP used 470 nm LED illumination, a 394/510/673 beam splitter, and a 412-433/501-547/617-758 triple band pass emission filter at 30% intensity with 150 ms exposure for samples in EMM and 50 ms exposure for samples in YES. Images were rescaled and cropped using custom Python scripts.

## RNA-seq data

Clément-Ziza et al. (2014) performed RNA-seq on the segregant library growing in EMM. We obtained a raw count matrix for an

unbiased search of correlations between gene expression and MLP formation (Clément-Ziza et al. 2014). Before the correlation analysis, raw count data was normalized using DESeq normalization (Anders and Huber 2010). For all additional omics data sources, see Supplementary Table 3.

## Finding shared upregulated genes across CKM deletions

After splitting segregants by their *srb11* haplotype, we performed differential expression analysis on the RNA-seq dataset from Clément-Ziza et al. (2014) using DESeq2 (Love et al. 2014). For further analysis, we used the top 100 upregulated protein-coding transcripts, by filtering for genes with log2 fold change (log2FC) > 0.5 and Benjamini–Hochberg adjusted *P*-value <0.05, and finding the entries with the top 100 lowest *P*-values. We compared this gene set against upregulated genes taken from the microarray dataset from Linder et al. (2008). Given that this dataset only contained sample means for each gene in each genotype, but no *P*-values, we simply took the genes with the top 100 largest log2FC values for both the *med12Δ* and *srb10Δ* genotypes. The intersection of the 3 gene sets then identified 15 genes which are upregulated in *srb11* truncation and *srb10Δ* and *med12Δ* genotypes.

## Finding overlap between genes upregulated upon CKM deletion and Mbx2 targets

Once we identified genes upregulated across the CKM deletion strains, we set out to compare this set with known targets of Mbx2. Kwon et al. (2012) have identified targets of Mbx2 by collecting microarray and ChIP-chip data. We considered genes with log2FC > 1 and Bonferroni-adjusted *P*-values <0.05 to be upregulated in the microarray dataset. Furthermore, as Kwon et al. (2012) have already performed quality control and filtered the ChIP-chip data for likely targets, we used every gene in that dataset. By finding the intersection of these 3 gene sets we identified 5 genes, including *mbx2*, which are upregulated in CKM deletion strains, likely through the activity of Mbx2. That *mbx2* itself is part of the gene list indicates that it binds its own promoter in a positive feedback loop.

## DNA sequencing data

Genotyping of 44 strains in the segregant library was done by Clément-Ziza et al. (2014). Briefly, they performed whole genome sequencing on the two parental strains with high coverage, and after calling short variants, they inferred the genotypes of segregants at each locus using bulk RNA-seq data. There remained 10 strains in the library that were not analyzed by Clément-Ziza et al. (2014), possibly because of their strong adhesive phenotype affecting downstream procedures.

We sequenced the remaining 10 strains in the library, and also the strain R4 as a control to compare our methods with that of Clément-Ziza et al. (2014). DNA extraction was done using spheroblasting followed by lysis, RNA and protein removal, and DNA extraction with the Qiagen Genomic-tip (20/G) protocol. Briefly, a 20 mL culture of cells was grown in YES at 32 °C and harvested by centrifugation (3,000 × *g* for 15 min at 4 °C). Cell walls were digested using lysing enzymes of *Trichoderma harizanum* dissolved in 50 mM citrate-phosphate buffer, pH 5.8, with 40 mM EDTA and 1.2M Sorbitol, and incubated for 1.5 h at 32 °C to generate spheroblasts. Cells were then centrifuged at 3,000 rcf for 10 min at 4 °C and supernatant was removed. Following this, the Qiagen Genomic-tip (20/G) protocol was followed, from page 37, step 8. Finally, high-quality DNA was isolated using isopropanol and ethanol precipitation. Library preparation and Illumina NovaSeq paired-end sequencing at well above 100× coverage was done by Azenta.

The resulting FASTQ files were checked for quality using FASTQC (Andrews 2010), following which we performed adapter trimming using Cutadapt (Martin 2011) using default parameters. The reads were mapped to the *S. pombe* reference genome using BWA MEM (Li 2013) using default parameters. The resulting alignments were then processed through the GATK short variant discovery pipeline (Poplin et al. 2018) using default parameters. In this pipeline, we used Base Quality Score Recalibration based on the .vcf file listing all known variants discovered by Jeffares et al. (2015). Although we collected haploid *S. pombe* samples, GATK assumed a diploid status during genotyping, which we used as a quality measure and discarded calls with heterozygous status, similarly to what was done before (Clément-Ziza et al. 2014).

We also accessed the original FASTQ files from the 2 parental strains (ENA accessions: ERX007392, ERX007395), and together with the newly sequenced 11 genomes (total of 13), we produced a variant call format file. This was then integrated with the genotype table from the supplemental material of Clément-Ziza et al. (2014). Uncalled variants where both the preceding and subsequent variants came from the same haplotype were imputed for each strain.

To evaluate the genotype calling pipeline, we compared the calls for the parental strains vs the calls made by Clément-Ziza et al. (2014). Using our pipeline on the sequencing data from Clément-Ziza et al. (2014) we saw that for JB50, our pipeline identified a wild-type genotype for 4,475 out of 4,481 (99.87%) variants found by Clement-Ziza et al. (46). For JB759, our pipeline identified the same alternative allele as Clement-Ziza et al. (46) 4,418 times out of the 4,481 (98.59%) variants. To compare our sequencing protocol and variant identification pipeline to that used in Clément-Ziza et al. (2014), we compared the calls for the segregant R4, for which we sequenced DNA and which had variants called based on RNA-sequencing data in Clément-Ziza et al. (2014). We found that the called SNPs differed at 26 loci out of 4,481 (0.58%), and concluded that both our computational genotyping and DNA sequencing pipeline was robust.

During the quality control step, we identified two previously unsequenced segregants in the segregant library that were duplicates of existing strains in the library. These strains had an extremely high overlap in variant calls of the two segregants R4 and R45 (>99% overlap), and the segregant R48 and the parental strain JB50 (>99% overlap). We therefore renamed R45 as R4_45 and R48 as JB50_48 and omitted them from further strain specific analysis. Because we now had two high-coverage replicates for R4, named R4 and R4_45, two for JB50 including the original sequence from Clément-Ziza et al. (2014) and newly sequenced JB50_48, and high coverage for the other newly sequenced samples, we used these sequences to call further short variants previously not reported in Clément-Ziza et al. (2014). The criteria were that these variants should be designated homozygous by GATK haplotype caller with different genotypes in the two parental strains, and that the variant should match between JB50_48 and the originally sequenced JB50 strain, as well as between R4 and R4_45. We called an additional 387 short variants, with an average length of 2.59 nucleotides for the newly called JB50 alleles and 2.88 nucleotides for the JB759 alleles. These stand in contrast with the variants called by Clément-Ziza et al. (2014) which were on average 1.22 and 1.15 nucleotides in length, meaning that we mostly identified indels while the previously called variants were mostly SNPs. In

the segregant strains genotyped only using RNA-seq by Clément-Ziza et al. (2014), these variants were imputed to match the haplotype of preceding and subsequent variants in the genome.

In the end, the sequencing efforts extended the dataset from 44 to 52 segregants, from 4,481 to 4,868 short variants, and from 685 to 816 haplotype blocks (Supplementary Table 4). Additionally, the raw sequencing data has been archived in the European Nucleotide Archive (www.ebi.ac.uk/ena/), with study accession PRJEB69522.

For the natural isolate library, we obtained genomic data from Jeffares et al. (2015) in a processed variant call format.

## QTL analysis

QTL analysis was done using the R-based RFQTL algorithm described in Michaelson et al. (2010) and Clément-Ziza et al. (2014), downloaded from http://cellnet-sb.cecad.uni-koeln.de/resources/qtl-mapping/. This method is based on the Random Forest machine learning algorithm. Briefly, short variants (SVs) and phenotypes are used to build decision trees, objects in which SVs with the highest explanatory power partition the phenotype data through a hierarchy of steps. Random subsets of data used for each decision tree give rise to a so-called random forest. There is always a "competing" collection of SVs being simultaneously considered, rather than a single variant as it is commonly the case for univariate statistical tests used for similar purposes (Michaelson et al. 2010). The hierarchical partitioning and the simultaneous consideration of multiple variants help to account for epistatic mechanisms and achieve higher fidelity QTL hits (Michaelson et al. 2010). Statistical significance is obtained as follows: first, an importance score ("selection frequency") is calculated for each SV on a small set of forests, and they are then compared with a null-distribution of importance scores coming from a large set of forests with bootstrapped data (Michaelson et al. 2010). For QTL analysis, we generated the importance scores from 100 forests with 100 trees each and created the null-distribution using 20,000 permutations of 100 forests with 100 trees each. The number of permutations was set such that $P$-values of genome-wide significance could be achieved given Bonferroni-correction for multiple-testing: n_permutations > n_haplotype_blocks/sig_threshold (= 803/0.05 = 16,060).

## Identifying variants causing a premature stop codon or frameshift

Identification of premature stop codons and frameshifts was done using a bespoke Python script that takes the reference genome .fasta file and genome annotation .gff3 file from PomBase, and an input of the variants of interest. After QTL analysis, this list comprised all 64 linked short variants showing statistical association with MLP formation. During analysis of CKM subunits, the query list comprised all known short variants (as identified in Jeffares et al. 2015) from the four CKM genes (srb10, srb11, med12, med13). Following the deletion library screen, the input list comprised all known short variants (as identified in Jeffares et al. 2015) in the 31 hit genes.

## Growth measurements

For liquid growth assays, three colonies for each strain were grown in 5 mL YES in culture tubes overnight. After measuring OD, a volume of cells sufficient to yield OD 0.1 in 1 mL was spun down and resuspended in 1 mL of either EMM or YES. Cells were then grown for 4 h in a deep well plate at 32C shaking at 1,000 RPM on a SI-200 Shaking incubator (Cole-Parmer). OD was then measured in a Spectrostar NANO (BMG labtech), and cells were again diluted to OD 0.1 in 1,000 µL. Then, 100µL of cells, including 3 to 4 technical replicates, were added to a nontreated flat bottom 96-well plate (CytoOne CC7672-7596) with a clear lid, the plate was placed in an incubator at 32 °C for 20 min to prevent condensation. The plate was then placed in the plate reader where OD600 was measured every 15 min (6 mm spiral sellscan, 0.1 settling time, 25 flashes) with continuous double orbital shaking at 700 rpm in between measurements and 30s of shaking before the first measurement.

## Gene set enrichment analysis

Gene set enrichment analysis (GSEA) was performed with the Bähler lab tool AnGeLi at http://bahlerweb.cs.ucl.ac.uk/cgi-bin/GLA/GLA_input (Bitton et al. 2015) on 2026 01 11. Gene sets included all categories and the significance threshold 0.01 was chosen with FDR correction for multiple-testing. GO terms that were deemed obsolete by the GO Consortium have been manually removed from the results.

## RT-qPCR

Cells were grown to the exponential phase (OD$_{600}$ ~ 0.5), and 15 ml aliquots were immediately spun down and stored at −80 °C. We extracted RNA using a standard hot-phenol protocol (Bähler and Wise 2017). We used Turbo DNase (Invitrogen) to digest the residual DNA and performed reverse transcription with the Superscript III kit and oligoDT primers (Invitrogen) according to the manufacturer's guidelines. We performed qPCR using Fast SYBR Green Master Mix (Applied Biosystems) on a QuantStudio 6 Flex instrument (Applied Biosystems) in fast cycling mode according to the manufacturer's instructions. Quantification of transcript abundance was done using a relative standard curve. For this, we pooled cDNA from samples expected to have the highest mbx2 concentration, and created a dilution series with 2×, 2/10×, 2/100× and 2/1,000× concentrations. We manually removed values for the 2× concentration, which showed strange amplification patterns for both mbx2 and act1 and would have led to inferred mbx2 fold change values an order of magnitude higher. For the standard curve and each of the three biological replicates, we measured 3 technical replicates.

## Results

### Regulators of MLP formation, but not cell-adhesion proteins, are conserved between fission and budding yeasts

To explore conservation in MLP formation between yeast clades, we identified all genes whose orthologs are annotated to an MLP-related GO term in at least one of C. albicans, S. cerevisiae and S. pombe (Materials and methods). We found that 371 of the annotated gene families (orthogroups) are conserved in all three species (Supplementary Fig. 2a). Intriguingly, however, only two orthogroups were functionally conserved, meaning that an ortholog was annotated to an MLP-related GO-term in all three species (Supplementary Fig. 2b). The first orthogroup includes the S. pombe transcription factor Mbx2 and its orthologs, known as Rlm1p in C. albicans, and the paralogs Rlm1p (Sariki et al. 2019) and Smp1p in S. cerevisiae. The second orthogroup included the kinase Amk2, and its orthologs, known as Kis1p in C. albicans, and the paralogs Gal83p and Sip2p in S. cerevisiae. Because the criteria for assigning phenotype annotations [e.g. via the FYPO (Harris et al. 2013) and via the S. cerevisiae and C. albicans model organism databases] are less strict than the criteria for assigning

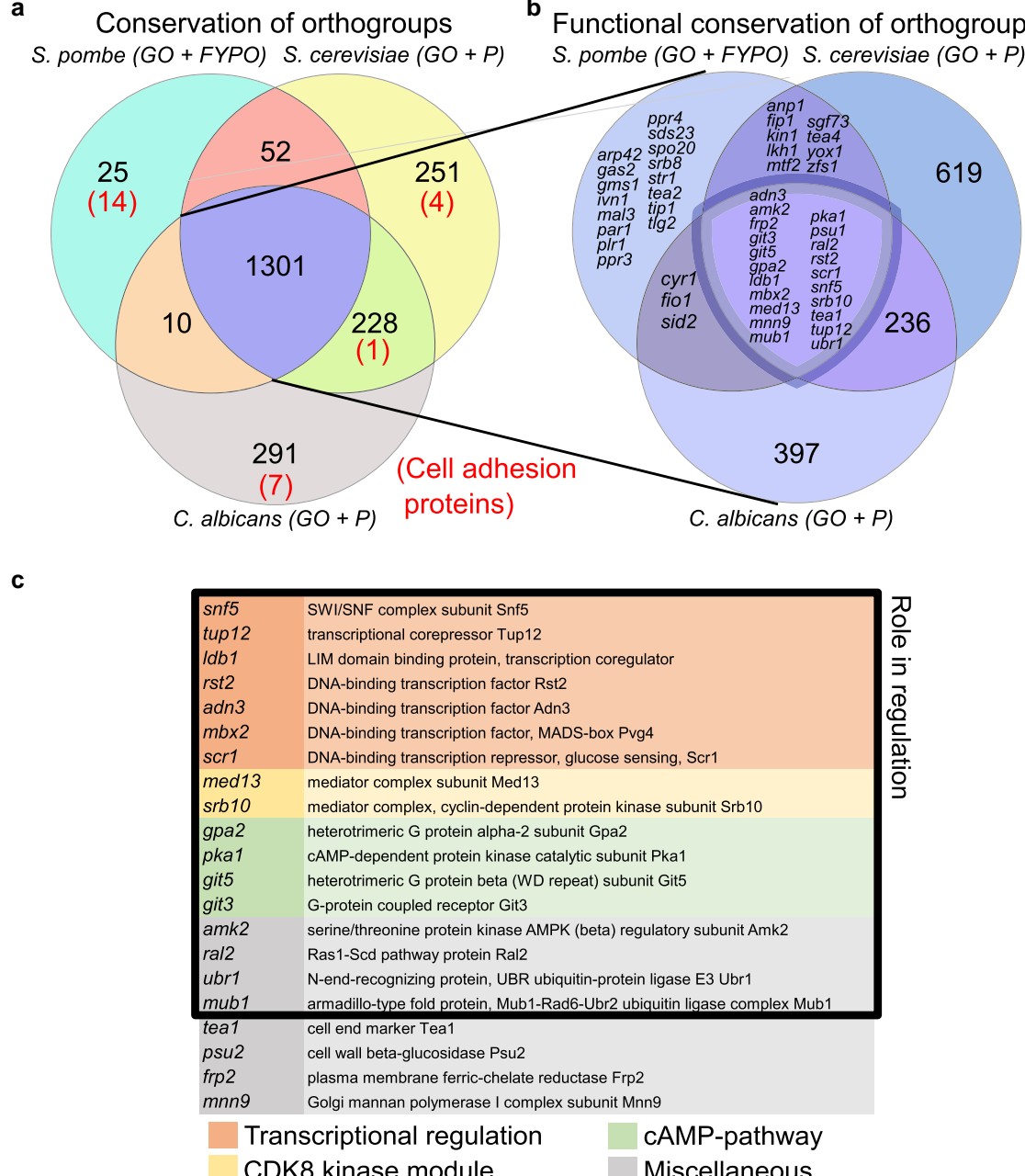

**Fig. 1.** Several regulators of MLP formation are conserved between fission and budding yeast, but cell-adhesion effector proteins are not. a) Venn diagram of numbers of orthogroups, in which at least one gene in the orthogroup is annotated in GO-terms or phenotypic data related to MLP formation in either *S. pombe*, *S. cerevisiae* or *C. albicans*. Numbers in parentheses indicate cell-adhesion proteins. b) Venn diagram of orthogroups that are conserved across the 3 species (i.e. the middle subset on A) asking whether they are also functionally conserved, i.e. contain at least one gene that is annotated in GO-terms or phenotypic data related to MLP formation in all three species. c) Functionally conserved genes colored by their broad functional category as indicated. FYPO, Fission Yeast Phenotype Ontology; GO, Gene Ontology; P, Phenotype annotations.

GO terms, phenotype annotations might uncover more conserved MLP-related genes. We therefore extended the analysis by incorporating annotations from phenotype databases (see Materials and methods). Interestingly, only 77 genes were annotated to relevant GO or phenotype terms in *S. pombe*, highlighting the stark knowledge gap in this area compared with *C. albicans* (1,103 genes) or *S. cerevisiae* (1,400 genes). Similar to GO-term annotations, in this wider set of MLP-related genes, most orthogroups related to MLPs in at least one of the three species were conserved (1,301/2,158) (Fig. 1a, Supplementary Table 5). However, only a small subset (21/1,301) of those genes was also functionally conserved as reflected by MLP-related annotations

in all three species (Fig. 1b). Besides Mbx2 and Amk2, these included additional proteins involved in transcriptional regulation, parts of the CKM of the Mediator, and members of the cAMP pathway (Fig. 1c).

Partial changes in biological pathways between related species, or biological circuit rewiring, have been well-documented (Rokas and Hittinger 2007; Booth et al. 2010). In several prominent cases, this rewiring has occurred at the regulatory level (Booth et al. 2010). However, our analysis suggests that, in the case of MLP formation, this rewiring appears to have happened at the level of downstream effectors, such as cell-adhesion proteins. Additional support for regulatory conservation comes from work

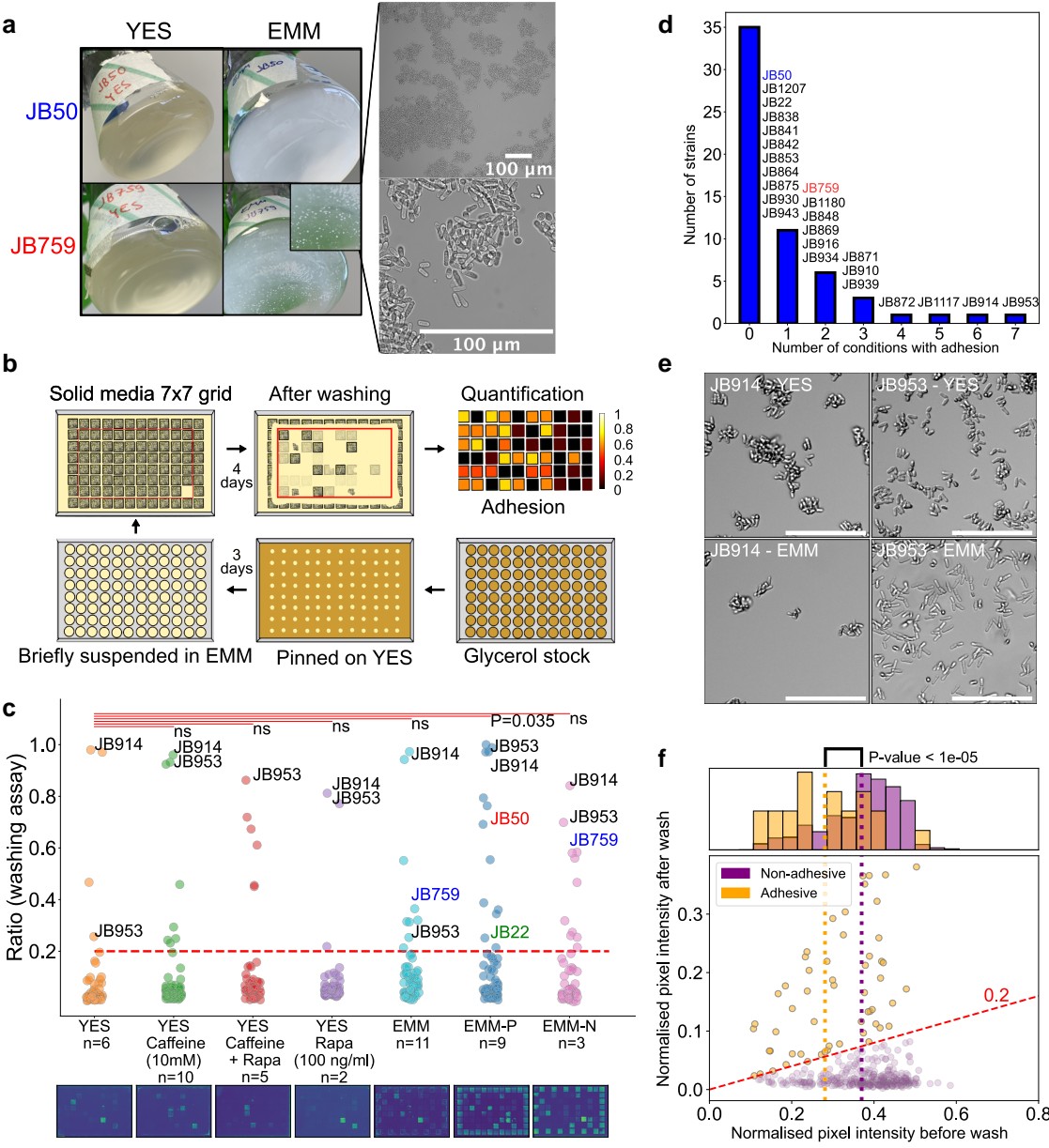

**Fig. 2.** MLP formation in *S. pombe* natural isolates varies with nutrient conditions and is associated with decreased growth. a) Left: Images of the initial observations on the standard laboratory strain, JB50, and the natural isolate, JB759, showing MLP formation of JB759 in EMM. Cells were grown at 32 °C for 2 d, shaking at 160 rpm. Right: Microscopy images at two magnifications of the JB759 strain grown in EMM for 2 d. b) Scheme of the high-throughput adhesion assay used to assess MLP formation in *S. pombe*. c) Strip plot of adhesion to agar across different conditions in the natural isolate library, with images of representative postwash agar plates below. Each dot represents the mean adhesion value for a given strain in a specific condition. The dashed line shows a cutoff for strong phenotypes (intensity after wash >0.2 times before wash). Each condition was compared with rich media (YES) with the null hypothesis that they do not increase MLP formation. *P*-values were obtained using a one-sided permutation-based T-test and Bonferroni correction. Comparisons were marked not significant (ns.) where the null hypothesis could not be rejected at significance threshold 0.05. Strains around the edges were not taken into account for any statistical analysis (see Materials and methods). The lab strain JB50 and natural isolate JB759 used in panel a), as well as the lab strain, JB22, and the highly adhesive strains JB914 and JB953, are highlighted in conditions in which they are adhesive. d) Histogram showing the number of unique strains forming MLPs in a given number of conditions. e) Panel of light microscopy images for JB953 and JB914 in EMM and YES, showing representative images of MLP-formation. Scale bars are 100 μm. Rest of the microscopy images from this experiment are displayed in Supplementary Fig. 4. Representative images of culture flasks are shown in Supplementary Fig. 5. f) Scatter plot of mean cell densities (measured in normalized inverse pixel intensity) before and after washing. Each dot represents one strain in one condition, points above the diagonal dashed line represent adhesive strains (ratio of before-wash to after-wash intensity >0.2) and points below the diagonal dashed line represent nonadhesive strains. The histogram shows the distribution of cell densities before washing as a proxy for growth. The left and right vertical dotted lines mark the mean prewash densities for the adhesive and non-adhesive populations respectively.

demonstrating that overexpression of the *S. pombe* transcription factor, Mbx2, can trigger MLP formation in *S. cerevisiae* (Matsuzawa et al. 2012). Furthermore, overexpression of the *S. cerevisiae* transcription factor, Flo8p (*S. pombe* orthologs Adn2 and

Adn3), can also trigger MLP formation in *S. pombe* (Matsuzawa et al. 2011).

In contrast, *S. pombe* cell-adhesion proteins, the downstream effectors of MLP formation, have little in common with other

fungal adhesins (Linder and Gustafsson 2008). *S. pombe* flocculins consist of repetitive beta-sheets and, commonly, a GLEYA or DIPSY sugar-binding domain at the C-terminus, unlike in other fungal adhesins where similar domains are N-terminal (Linder and Gustafsson 2008). While the GLEYA domain is similar to the lectin-like ligand-binding domain of certain *S. cerevisiae* flocculins (Linder and Gustafsson 2008), the DIPSY domain has only been identified in species of the *Taphrymycotina* lineage. The dominant flocculin, Gsf2, contains neither of these domains and appears to be unique to *S. pombe,* even within the *Taphrymycotina* lineage, with no detected *S. japonicus* ortholog (Harris et al. 2022; Rutherford et al. 2022). To verify that *S. pombe* flocculins are indeed poorly conserved in other yeast species, as suggested by the database annotations, we performed sequence- and structure-based queries using BLAST-P (Altschul et al. 1997) and Foldseek (van Kempen et al. 2024), respectively, for *S. pombe* cell-adhesion proteins against all proteins in *S. cerevisiae* or *C. albicans* (Materials and methods). Both sequence- (*C. albicans*, $P = 1.8E-5$; *S. cerevisiae*, $P = 5.2E-7$) and structure-based alignment scores (*C. albicans*, $P = 0.02$; *S. cerevisiae*, $P = 1.3E-5$) were significantly lower for *S. pombe* flocculins compared with queries from a random set of 50 conserved proteins (Supplementary Fig. 2c, Supplementary Table 2), further supporting our observation that cell-adhesion proteins in *S. pombe* are either lineage-specific or weakly conserved.

Taken together, our bioinformatic analysis suggests that, aside from a few key regulatory proteins, most genes involved in MLP formation are not functionally conserved between fission yeast and budding yeast. Some of this discrepancy may be attributed to annotation bias, given the less work done on *S. pombe* in this subject. However, the presence of genes that are annotated as contributing to MLP-formation in *S. pombe* but not annotated as such in budding yeast argues for true divergence. Additionally, cell-adhesion proteins seem to differ greatly between fission yeast and budding yeast. Lastly, the much smaller number of genes annotated to MLP formation in *S. pombe* highlights that these phenotypes are understudied in fission yeast.

## Multicellular-like phenotypes depend on environmental context

We observed that the *S. pombe* natural isolate JB759 (Y0036) sticks to the side of glass flasks and forms clumps in liquid minimal media (EMM) but not in rich media (YES) (Fig. 2a). To explore the natural variation in MLP formation across strains and conditions, we developed high-throughput methods to assay flocculation and adhesion to agar (Materials and methods) and applied them to a collection of 57 genetically diverse natural isolates (Fig. 2b and c, Supplementary Fig. 3a) (Jeffares et al. 2015). The two phenotypes were strongly positively correlated with each other ($r = 0.8$, $P = 4E-14$, Supplementary Fig. 3a). This result points to a shared mechanism underlying the two distinct MLPs.

Extending the initial observation, depending on the strain, the penetrance of surface-adhesion phenotypes varied across different nutrient and drug conditions (Fig. 2c, Supplementary Table 6). Compared with adhesion to YES plates, only phosphate starvation (EMM-P) led to significantly increased mean adhesion levels ($P = 0.035$, one-tailed permutation-based T-test). Although the other 6 conditions did not lead to significantly changed mean adhesion levels in the strain collection, between 3 and 13 strains in each condition passed the threshold for a strong adhesion phenotype (defined using an elbow plot, Supplementary Fig. 3b). Compared with the 4 strains that passed that threshold in YES, there were more adhesive strains in EMM-P ($n = 13$),

nitrogen starvation (EMM-N) ($n = 11$), EMM ($n = 9$), YES with caffeine ($n = 8$) and YES with caffeine and rapamycin ($n = 6$). Out of the 57 natural isolates, 24 strains (42%) showed a strong adhesion phenotype in at least 1 condition. For most strains, such strong adhesion was limited to 1 or 2 conditions, but 2 strains, JB914 and JB953, showed strong adhesion in 6 or all 7 conditions tested (Fig. 2d, representative images in EMM and YES shown in Fig. 2e and Supplementary Figs. 4 and 5). JB914 and JB953 had been isolated from molasses in Jamaica and the exudate of Eucalyptus in Australia, respectively (Jeffares et al. 2015). Strikingly, even the 2 lab strains, JB22 and JB50, showed strong adhesion under phosphate starvation (EMM-P; Supplementary Fig. 3c), while the adhesion level of JB759 was below the threshold in that condition (Fig. 2c). Indeed, RNA-seq from (Garg et al. 2023) revealed that the flocculation-related transcription factor gene *mbx2* and downstream cell-adhesion genes are upregulated with time under phosphate starvation (Supplementary Fig. 3d). Lastly, comparing measurements across all conditions, we found that strains exhibiting strong adhesion generally featured lower colony density before washing (0.28 vs 0.37, measured as normalized inverse pixel intensity), which may reflect a decreased growth rate (Permutation-based T-test: $P < 1E-5$, Fig. 2f).

## Truncation of the *srb11* gene causes MLP formation in an adhesive natural isolate

Motivated by the findings above, we dissected the genetic mechanism underpinning MLP formation on EMM in the JB759 strain, isolated from wine in South Africa (Brown et al. 2011). We used an existing segregant library generated by crossing the JB759 strain with the lab strain, JB50 (Clément-Ziza et al. 2014) (Fig. 3a). Flocculation and adhesion to agar on EMM were highly correlated with each other in this library (Supplementary Fig. 6a, $r = 0.88$, $P = 2E-20$). Notably, adhesion occurred only on EMM and not on YES media (Permutation-based T-test, $P < 1E-6$; Fig. 3b).

An RNA-seq dataset for this segregant library, measured during exponential growth in EMM, was previously published (Clément-Ziza et al. 2014). We performed an unbiased search for correlations between gene expression in that dataset and flocculation, as measured using the filtering assay (Materials and methods). We found 242 genes (5% false discovery rate), of which 138 were protein-coding, to be significantly associated with this phenotype (Supplementary Table 7). The four transcripts showing the highest correlation with flocculation encoded the transcription factor Mbx2 and the flocculins Pfl8, Pfl3 and Pfl7, while the transcript encoding the dominant flocculin Gsf2 was also highly correlated (Fig. 3c). Accordingly, *mbx2* gene expression showed a strong association with flocculin gene expression (Supplementary Fig. 6b).

Overexpression of *mbx2* is sufficient to cause flocculation in EMM (Kwon et al. 2012). We tested whether *mbx2* overexpression also causes agar adhesion, as measured by our assays, in both EMM and YES. To this end, we engineered *mbx2* overexpression strains in *h-* and *h90* backgrounds. These strains exhibited flocculation in liquid media (both EMM and YES), while wild-type controls and GFP-overexpression strains did not (Fig. 3d, Supplementary Fig. 7). The flocculation phenotype was so strong that our original adhesion assay protocol required modification for this experiment, including resuspension in 10 mM EDTA before repinning (Materials and methods). We observed strong adhesion for the *mbx2* overexpression strains (Fig. 3d), but not for control strains in YES. We could not check adhesion in EMM for

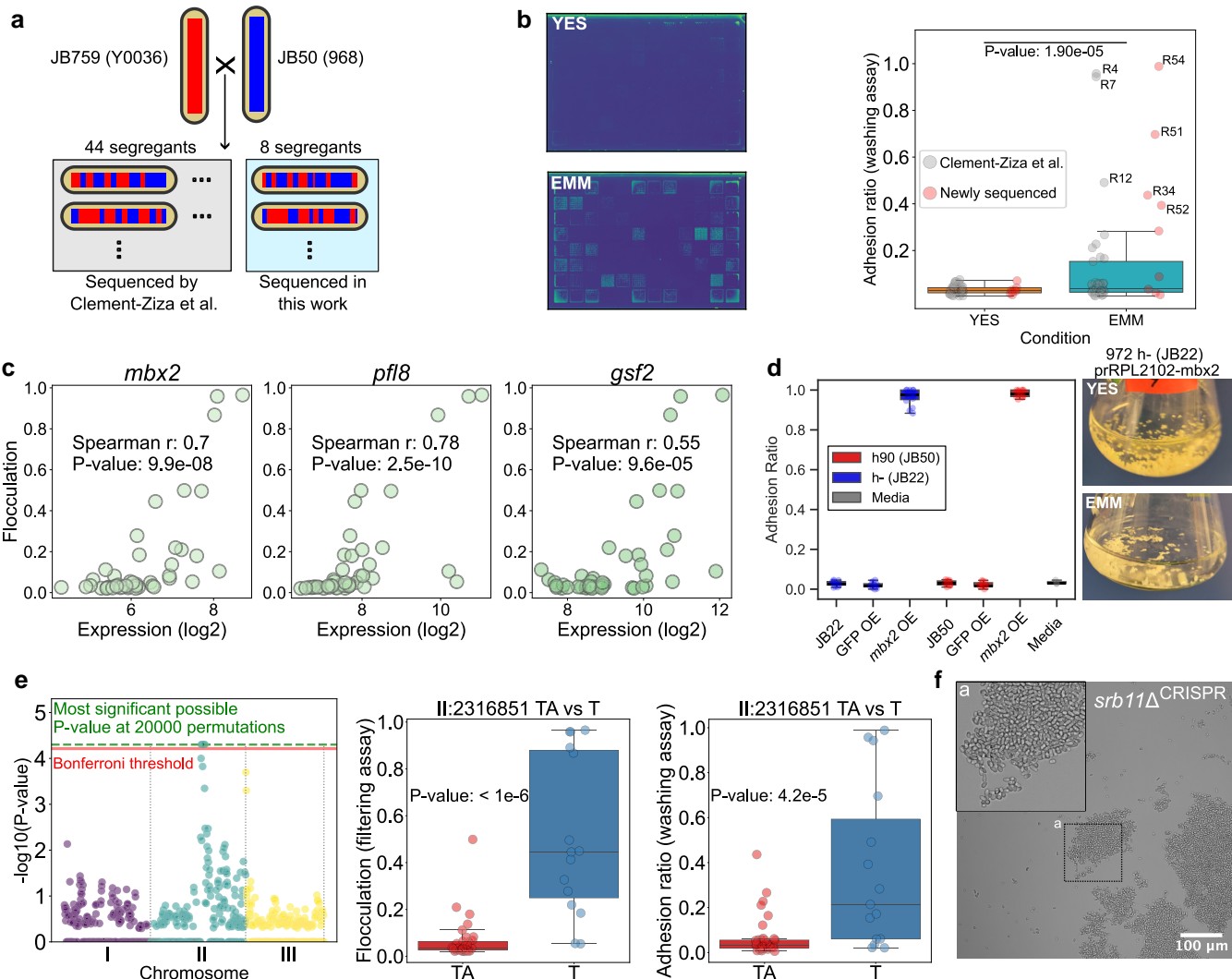

**Fig. 3.** MLP formation in JB759 is driven by *mbx2* expression and associated with a single-nucleotide deletion on chromosome II. a) Scheme for the JB759xJB50 segregant library. Red and blue stripes indicate genomic recombination resulting from meiosis. SNPs for the strains inside the gray box were previously identified (Clément-Ziza et al. 2014) and genome sequencing data for the strains in the light blue box were generated in this work (Materials and methods). b) (Left) Example plate scans of segregants grown on EMM or YES after washing, shown in viridis colormap. (Right) Adhesion to agar of segregant strains on EMM (mean of 10 replicates) compared with YES (mean of 2 replicates), along with significance of the difference obtained using permutation-based T-test. c) Correlations of *mbx2* and flocculin gene expression with flocculation in EMM. Each dot represents a strain from the segregant library. d) (Left) Box plot of adhesion measurements comparing adhesion measurements from control, GFP-overexpression and *mbx2* overexpression strains in both JB22 and JB50 backgrounds, as well as measurements for empty media (Materials and methods). (Right) Representative images of the flocculating *mbx2* overexpression strain in YES and EMM. Rest of the flask images are found in Supplementary Fig. 7B. e) (Left) Manhattan plot of QTL analysis results for flocculation in EMM. The solid line shows the Bonferroni threshold, while the dashed line shows the highest possible significance achievable using 20,000 permutations. (Middle) Candidate variant II:2316851 TA > T is associated with increased flocculation in EMM (filtering assay, mean of 3 replicates). (Right) Candidate variant II:2316851 TA > T is associated with increased adhesion on EMM (washing assay). *P*-values determined using a permutation-based T-test with 1E + 6 permutations. f) Microscopy image of *srb11Δ*^CRISPR in EMM showing flocculation. The inset shows a close-up of the region highlighted in box "a".

those strains because cells pinned on EMM did not survive resuspension in EDTA.

To identify genetic determinants of the variation in flocculation amongst the segregants, we mapped Quantitative Trait Loci (QTL) using genotype data from 44 of the 52 segregants in the library (56). To increase the statistical robustness of our dataset, we sequenced the remaining 8 nonredundant strains from the segregant library, including several highly flocculant strains, for which SNPs were not previously identified (Materials and methods). This enabled us to pinpoint a genomic region of 65 short variants that was strongly associated with both flocculation and adhesion to agar (Fig. 3e). These variants included 26 open reading

frames with 13 variants resulting in a changed codon. One of these variants introduced a premature stop codon through a frameshift in the gene *srb11*, encoding cyclin C. To validate that disruption of *srb11* causes MLP formation, we generated a CRISPR deletion strain (*srb11Δ*^CRISPR), which indeed exhibited flocculation and strong adhesion to agar, on EMM (Fig. 3f).

The identified variant in *srb11* leads to the production of a truncated (77/228 amino acids) form of Cyclin C (Fig. 4a and b). Cyclin C (Srb11), together with cyclin-dependent kinase 8 (Srb10) and Mediator subunits Med12/Srb8 and Med13/Srb9, is part of the CKM of the Mediator complex (Fig. 4b). The CKM can impede transcription by inhibiting the interaction between the core Mediator

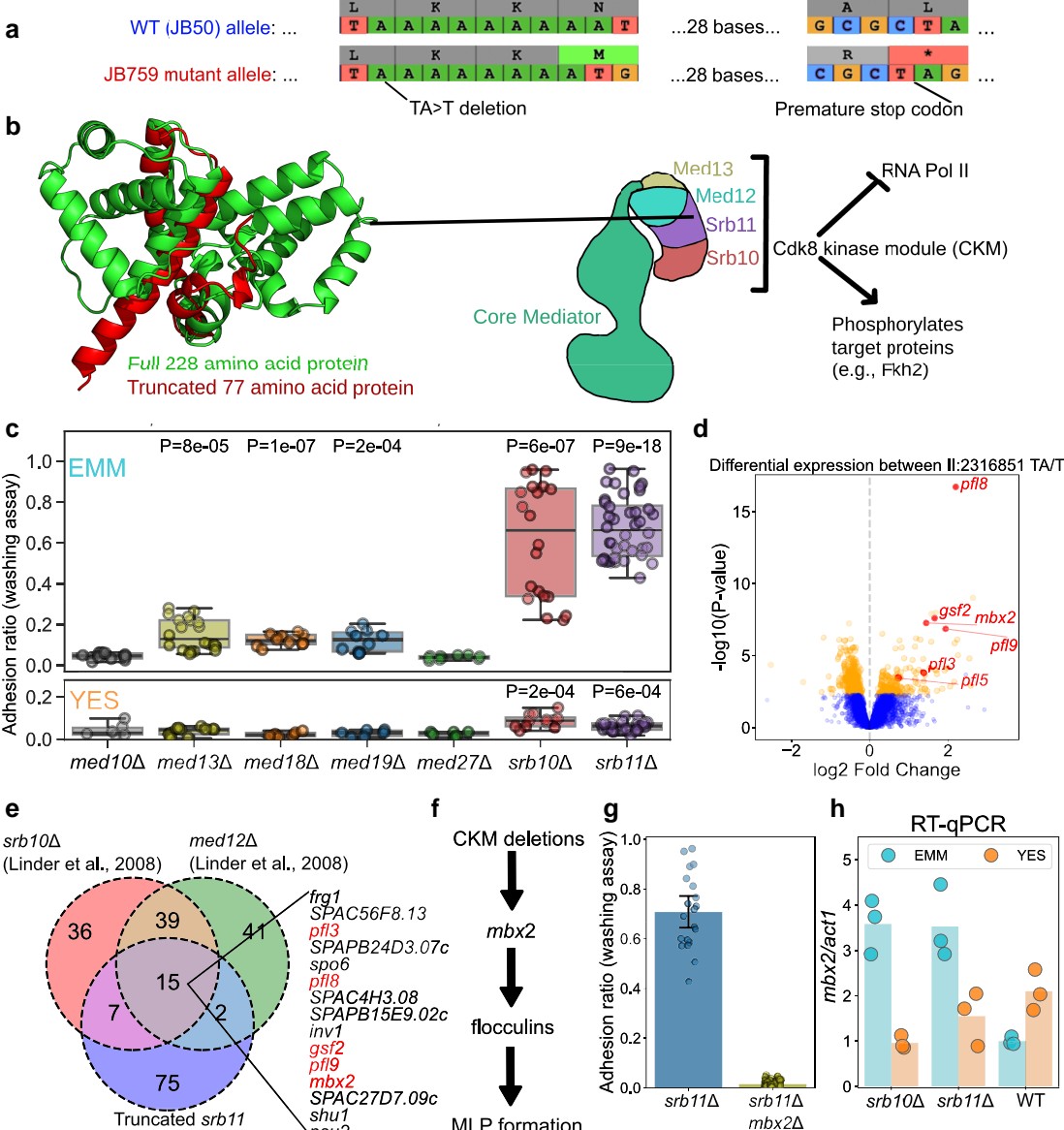

**Fig. 4.** Cdk8 kinase module deletions upregulate *mbx2* in minimal media, but not in rich media, and lead to MLP formation. a) Scheme showing how a single-nucleotide deletion leads to frameshift and premature stop-codon in *srb11*. b) Full Srb11 structure (green) compared with the truncated Srb11 structure (red) as predicted using Colabfold (Mirdita et al. 2022). Scheme on the right shows Srb11 in the context of the CKM of the Mediator complex. The structure was sketched based on structural data (Tsai et al. 2013), and functional roles were summarized based on (Szilagyi et al. 2012; Tsai et al. 2013). c) Box plot showing adhesion values from Mediator gene-deletion strains from the genome-wide prototrophic deletion library on EMM and YES as indicated. Each dot represents a replicate. P-value represents comparison of each strain against the least adhesive strain in the assay (*med27Δ*) using a T-test. d) Differential expression analysis of segregant strains split on the II:2316851 TA > T single-nucleotide deletion. Fold-change values and *P*-values were obtained from DESeq2 (Love et al. 2014). Flocculins and *mbx2* are highlighted with labels. e) Overlap of upregulated genes in three CKM mutants based on our *srb11* data and data from Linder et al. (2008). f) Simplified model for how CKM deletions lead to MLP formation. g) Adhesion measurements for the *srb11Δ* strain obtained from the deletion collection, and its derived strain after *mbx2* knock-out with CRISPR. Each dot represents a replicate measurement. Error bars represent the 95% confidence interval for the mean. h) RT-qPCR showing *mbx2* expression in JB22 (WT), *srb10Δ*::*Kan*, *srb11Δ* strains obtained from the genome-wide prototrophic deletion collection in EMM or YES. Expression is normalized relative to *act1* expression. Height of each bar reflects the mean of three biological replicates which are indicated by dots.

and RNA polymerase II (Elmlund et al. 2006; Zhu et al. 2006; Tsai et al. 2013), and phosphorylates various target proteins (Nelson et al. 2003; Raithatha et al. 2012; Szilagyi et al. 2012; Banyai et al. 2014; Hollomon et al. 2022). Previously it was found that deletion of *srb10*, *med12* and *med13* causes flocculation (Samuelsen et al. 2003; Linder et al. 2008), which was supported by microarray data showing increased expression of flocculins in these mutants (Linder et al. 2008).

To better understand the role of the Mediator complex in cell adhesion, we studied deletion mutants of all Mediator subunits available in the genome-wide prototrophic gene-deletion library (Kim et al. 2010; Malecki and Bähler 2016) (Supplementary Table 8). Consistent with results from our *srb11Δ*[CRISPR] strain and prior work (Watson and Davey 1997; Linder et al. 2008), we found that the *srb11Δ* and *srb10Δ* strains flocculated (Supplementary Fig. 6d) and exhibited adhesion to agar, particularly on EMM (Fig. 4c).

Deleting *med13*, another component of the CKM, showed a milder phenotype compared with the extreme adhesion of *srb10Δ* and *srb11Δ* cells (Fig. 4c), while the fourth component of the CKM, *med12*, was not represented in the deletion library. Deletion of the core Mediator genes *med19/rox3* and *med18* also resulted in mild adhesion phenotypes, while deletion of *med10/nut2* and *med27/pmc3* did not (Fig. 4c). We then tested whether flocculation or adhesion to agar in the CKM mutant strains depends on media composition. Interestingly, the core Mediator mutants and *med13Δ* only showed their adhesion phenotype on EMM, while *srb10Δ* and *srb11Δ* showed a strong phenotype on EMM, as well as a very mild adhesion phenotype on YES (Fig. 4c). To confirm that these results did not result from possible artifacts in the deletion library, we created fresh deletion mutants for *med10*, *med13*, *med18*, *med19*, *med27*, *srb10*, as well as *med12*, which was missing from the deletion library (Supplementary Fig. 6e). The dynamic range of our assay was smaller for this experiment, possibly due to variability during the washing step, and *med12Δ*, *med13Δ*, *med18Δ*, and *med27Δ* cells did not show a significant increase in adhesion. However, the *med10Δ* deletion strain showed a mild effect in EMM, and, importantly, we confirmed that *srb10* and *srb11* have strikingly different adhesion phenotypes compared with other Mediator subunit genes and that the phenotype is stronger in EMM (Supplementary Fig. 6e).

Next, to identify upregulated genes in strains containing the *srb11* truncation, we returned to the RNA-seq dataset on the segregant library (Clément-Ziza et al. 2014). Here, we split the strains by their *srb11* haplotype, then performed differential expression analysis (Fig. 4d, Supplementary Table 9). The genes associated with the *srb11* truncation were compared with genes differentially expressed relative to wild-type cells in *srb10Δ* and *med12Δ* strains grown in EMM (Linder et al. 2008). We found that *mbx2* and four flocculin genes (*pfl3*, *pfl8*, *gsf2*, and *pfl9*) were upregulated in all 3 datasets, together with 10 other genes (Fig. 4e). One of these 10 genes was *inv1*, encoding the external sucrose invertase, and whose ortholog in *S. cerevisiae*, *SUC2*, can enable nutrient sharing among cell aggregates (Koschwanez et al. 2013; Chow et al. 2019a). These genes also included the cell-surface heme-acquisition gene *shu1*, which causes flocculation and filamentous growth when overexpressed (Su et al. 2018). These results suggested a model in which CKM deletions lead to MLP formation through upregulation of *mbx2* which in turn leads to the upregulation of flocculins (Fig. 4f).

To validate this model, we asked how many of the genes upregulated in the CKM deletion mutants overlap with genes known to be activated by Mbx2. Based on microarray and ChIP-chip data (Kwon et al. 2012), the four flocculins and *inv1* are indeed regulated by Mbx2 (Supplementary Fig. 8a). To test whether MLP formation in the *srb11Δ* strain requires *mbx2*, we created a *srb11Δmbx2Δ* double deletion using CRISPR-Cas9 gene-editing (Rodríguez-López et al. 2016). The *srb11Δmbx2Δ* strain did not exhibit flocculation in liquid media (Supplementary Fig. 8b) or adhesion to agar (Fig. 4g). We conclude that Mbx2 is essential for the MLP phenotype seen in *srb11Δ* cells.

Given the model that the *srb11* truncation upregulates *mbx2*, we wondered why the JB759, *srb10Δ* and *srb11Δ* strains only exhibited strong MLP formation in EMM. The transcriptomics data from the segregant library (Clément-Ziza et al. 2014) and the CKM deletion strains (Linder et al. 2008) both came from cells grown in EMM. We therefore tested whether the upregulation of *mbx2* in CKM deletion strains is specific to EMM, like MLP formation. To this end, we performed RT-qPCR in wild type (JB50) and *srb10Δ* and *srb11Δ* strains from the deletion library grown in EMM or

YES to measure the expression of *mbx2*. Upregulation of *mbx2* in these two CKM mutants was indeed exclusive to EMM (Fig. 4h). Upregulation of *mbx2* in CKM mutants in EMM (3.6-fold for *srb10Δ* and 3.5-fold for *srb11Δ*) was of a similar magnitude to the increase observed in microarray data from an *srb10Δ* deletion strain (9.3-fold, Linder et al. 2008) and in bulk segregant data averaged over *srb11* truncation haplotypes (Fig. 4d, 2.7-fold, Clément-Ziza et al. 2014).

The protein components of the *S. pombe* CKM physically interact (Samuelsen et al. 2003; Banyai et al. 2017), and their deletion typically generates similar phenotypes and transcriptomic profiles in *S. pombe* (Linder and Gustafsson 2008) and *S. cerevisiae* (van de Peppel et al. 2005). However, here we show that under our growth conditions, deletion of the kinase or cyclin gene (*srb10* and *srb11*) leads to strikingly different adhesion phenotypes than deletion of the other two subunits (*med12* and *med13*). To see whether this is true for other phenotypes, we analyzed data from Rodríguez-López et al. (2023), who measured sensitivity and resistance phenotypes of deletion strains in 131 conditions. In those data, deletion of *med13* and *srb11* resulted in different phenotypes across a range of growth conditions. In terms of these phenotypes, *med13Δ* is only in the 21st percentile for similarity to *srb11Δ* (Supplementary Fig. 8c). Interestingly, the top 10 deletion strains that were phenotypically most similar to *srb11Δ* included *ace2Δ* and *cbf11Δ*, both of which have been found to trigger MLP formation (Linder et al. 2008; Převorovský et al. 2009a) (Supplementary Fig. 8c).

The only well-documented physical interaction of Srb10/Srb11 in *S. pombe* is the stabilization of the transcription factor Fkh2 by phosphorylation (Szilagyi et al. 2012; Banyai et al. 2014). We therefore checked whether loss of *fkh2*, similarly to loss of *srb10/srb11*, leads to *mbx2* upregulation. ChIP-seq data indicated that Fkh2 does not bind to the *mbx2* promoter (Garg et al. 2015). Furthermore, re-analysis of microarray data from Szilagyi et al. (2012) and Garg et al. (2015) indicated that deletion of *fkh2* does not increase levels of *mbx2* (Supplementary Fig. 8d, e). The mechanism that inhibits the upregulation of *mbx2* in YES also remains unknown, as *gsf1*, which codes for the only known repressor of *mbx2*, is not significantly upregulated in YES relative to EMM (Atkinson et al. 2018, Supplementary Fig. 8f).

Finally, we checked whether variants similar to the *srb11* truncation appear in any other CKM genes within the natural isolate library using the genotype data from Jeffares et al. (2015). Besides JB759, the parental strain of the segregant library, we only found the same *srb11* frameshift mutation in the unrelated, strongly flocculant strain, JB914. Though JB759 and JB914 differed at 70,640 out of 73,602 (96.0%) short variants called using the wild-type strain as reference (Jeffares et al. 2015) (Supplementary Fig. 9a), in a ±20 Kb window near the frameshift, they differed in only 3 out of 22 (13.6%) variants (Supplementary Fig. 9b). This finding raises the possibility that the frameshift mutation did not appear twice independently, but perhaps is a result of recombination. We find that none of the strains closely related to JB759, in terms of either global genotype (Jeffares et al. 2015) and within the ±20 Kb window around the frameshift mutation, shared the mutation or formed MLPs. This result provides further evidence that MLP formation segregates with the *srb11* short variant amongst natural isolates (Supplementary Fig. 9c). Interestingly, JB914 exhibited strong adhesion to agar (and flocculation) on both EMM and YES (Fig. 2c and d), indicating the likely presence of additional variants in the pathway inhibiting MLP formation in YES.

In summary, we used a segregant library to dissect the genetic determinants of MLP formation on EMM in the JB759 strain. We

found a single-nucleotide deletion that leads to a truncation of Srb11 to be associated with MLP formation on EMM. We further determined the effect of this truncation to be the EMM-specific upregulation of the transcription factor gene *mbx2*. Upregulation of *mbx2* in turn leads to the upregulation of cell-adhesion genes, which mediate MLP formation. Using an *mbx2* overexpression strain and a *srb11Δmbx2Δ* double mutant, we showed that upregulation of *mbx2* is both necessary and sufficient to explain MLP formation in the *srb11* mutant JB759. Lastly, we showed that the *srb11* truncation variant segregates with MLP formation in the natural isolate collection as well.

## Novel players in MLP formation on minimal media

The premature stop codon in *srb11* only accounts for the phenotype of two (JB759 and JB914) of the 9 natural isolates that exhibited MLP formation on EMM. Therefore, to identify further possible genetic causes of agar adhesion on EMM, we screened the genome-wide prototrophic gene-deletion library (Kim et al. 2010; Malecki and Bähler 2016) and a lncRNA deletion library (Rodriguez-Lopez et al. 2022). We performed one round of the adhesion-to-agar assay on EMM for 3,721 unique deletion strains (a total of 4,327 strains including replicates). While not every strain grew on YES after initial inoculation, or grew on EMM during the screen, we successfully assayed a total of 3,628 strains (Supplementary Fig. 10a). The *srb10Δ* and *srb11Δ* strains failed to be inoculated for this initial screen, likely due to aggregation at the bottom of the 96-well plate. This indicates that some other true positive genes related to MLP formation may have been missed in our initial screen.

Given that our measure for adhesion is the fraction of cells remaining after washing (after/before), we considered that strains with minimal growth before washing (denominator) might appear to have higher ratios, despite only negligible intensity values after washing (attributed to measurement error rather than biological signal). However, a scatter plot of all measurements argues against such systematic bias, as adhesive strains covered a wide range of prewash growth values (Fig. 5a). Still, the adhesive deletion strains exhibited a decreased growth phenotype on average (Fig. 5a, Permutation-based T-test, $P < 10E-5$), but we attribute this to a biological effect similar to that seen in the natural isolates (Fig. 2f). Based on the assay, deletion strains of protein-coding genes with adhesion values in the top 5th percentile were chosen for functional enrichment analysis (Supplementary Fig. 10b). Mutants showing the strongest adhesion phenotypes were enriched for ribosomal protein genes and for other genes associated with slow-growth phenotypes (Fig. 5b, Supplementary Tables 10 and 11).

To validate these findings, we narrowed down our search to the most adhesive strains. By arranging them in the middle 60 spots of three 96-well plates and including a positive control (strongly adhesive JB914 strain) and negative controls (nonadhesive deletion strains and the lab strain JB50, as well as an empty square), we were able to quantify the adhesion of these strains more precisely. In this confirmation step, we found 31 high-confidence hits (Supplementary Table 12), defined as deletion strains where, in at least 5 repeats, cell density before and after washing (>0.1 normalized pixel intensity before and >0.05 after) was sufficient to allow robust quantification of adhesion, and the adhesion ratio was >0.086, the 95th percentile cutoff for our initial screen. Interestingly, except for the *sre2* mutant, all adhesion phenotypes were either milder or not present on YES (Fig. 5c, Supplementary Fig. 11).

Out of these high-confidence hits, *sre2*, encoding a sterol regulatory element binding transcription factor (Kwon et al. 2012), *rpl2102*, encoding a part of the large ribosomal subunit (Li et al. 2013), and *med18*, encoding a component of the Mediator head domain (Grallert et al. 1999; Linder et al. 2008), have been previously implicated in cell adhesion or filamentous growth. We found three lncRNA deletions to exhibit adhesion, but at least 2 of these may affect protein-coding genes: SPNCRNA.1234 entirely overlaps with the gene *nmt1*, while SPNCRNA.781 is near the promoter of *hsr1*, a transcription factor which has recently been linked to flocculation (Ohsawa et al. 2024). However, *nmt1* and *hsr1* deletion mutants were present in our screen, and did not show strong adhesion. The third lncRNA, SPNCRNA.900, is located between two genes, *glt1* and *eme1*, both of which appears to be obviously related to MLP formation, and therefore could be a trans-acting noncoding RNA that influences MLP formation.

We then asked whether these hits might belong to the same pathway or represent distinct triggers for MLP formation. To answer this, we returned to the large-scale phenotypic dataset from Rodríguez-López et al. (2023) (analyzed for Supplementary Supp Fig. 8c), where the authors identified 8 broad phenotypic clusters of deletion strains. Our high-confidence hits were present in 6 out of 8 phenotypic clusters (Supplementary Fig. 12a). This suggests that while deletions of these genes all lead to MLP formation on EMM, they represent different pathways, and their deletions lead to different phenotypes across conditions. The previous large-scale phenotypic study also classified deletion strains as slow- or fast-growing in solid EMM media compared with wild-type. 18 of the 28 protein-coding hits from our screen were found in the slow-growing category ($P = 1.35E-6$, Fisher's exact test), and none in the fast-growing category (Supplementary Fig. 12b). This result reinforces our previous conclusions linking decreased growth and MLP formation.

We looked at whether these hits are also functionally conserved in *C. albicans* and *S. cerevisiae*. Four protein-coding genes (SPAC607.02c, *sre2*, *meu27*, *for3*) do not have an ortholog in the two budding yeast species. From the orthogroups that are genetically conserved, ribosomal gene deletions only affected MLP formation in *S. pombe*, while some orthogroups contained genes related to MLP formation in *C. albicans* (*csk1*, *hst4*, *kgd2*, *shm1*) or *S. cerevisiae* (*med18*, *mmp1*, *puf4*, *tom70*). Only two genes, encoding the transcription factor *fkh2* and the Mlu1-binding factor (MBF) transcription factor complex subunit *res1*, seem to have a conserved role across all three species in the regulation of MLP formation (Hollenhorst et al. 2000; Zhu et al. 2000; Bensen et al. 2002) (Fig. 5d).

Finally, we checked whether variants in these hit genes appeared in any of the natural isolates, but we did not identify a severe frameshift mutation in any of the 31 genes (Supplementary Table 13). Given that such mutations would reduce growth efficiency, it is not surprising that they are absent in natural isolate genomes. This, however, raises the question of why the *srb11* null-mutation would appear in two natural isolates as a driver of MLP formation. To investigate this question, we examined the tradeoff in MLP formation vs growth efficiency across all of our measurements, revealing that *srb10* and *srb11* deletions present an ideal combination of strong adhesion (2nd and 5th most adhesive) and growth efficiency (both above 5th percentile of all nonadhesive deletion strains) compared with other adhesive deletion strains (Fig. 5e). To further test this hypothesis, we obtained growth curves for WT (JB22), *srb10Δ*, *srb11Δ*, as well as for a subset of deletion strains that exhibited high levels of adhesion (*mus7Δ*, *rpa12Δ*, *kgd2Δ*, and *tlg2Δ*) in liquid EMM and YES. This analysis

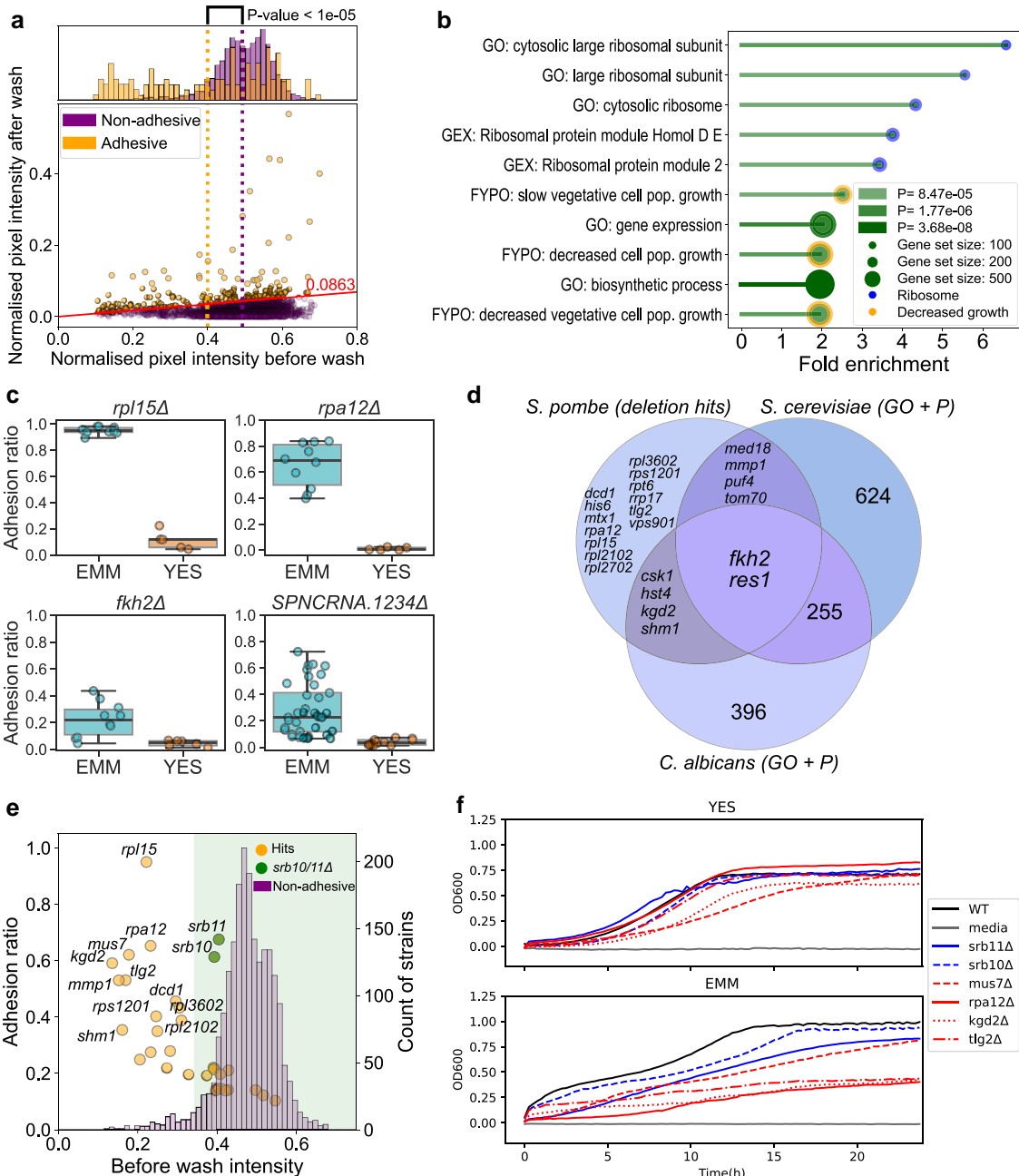

**Fig. 5.** Deletion library screen identified 31 genes associated with MLP formation on minimal media. a) Scatter plot of mean cell densities before and after washing. Each dot represents a deletion strain, and colors represent adhesive and nonadhesive strains as indicated. The diagonal line represents the cutoff at the 95th percentile of adhesion values (Supplementary Fig. 10b). The histogram shows the distribution of cell densities before washing as a proxy for growth. The left and right dotted vertical lines mark the mean prewash intensity values for the adhesive and non-adhesive populations respectively. The P-value was determined using a permutation-based T-test with $1E+5$ permutations. b) Bar plot showing fold enrichment for the top 10 most significantly enriched processes from AnGeLi (Bitton et al. 2015). Blue circles indicate terms associated with ribosomes, and orange circles indicate terms associated with decreased growth. Gene sets were sorted based on P-values, with increasing color intensity representing increasing −log10(P-value). The size of the circle at the end of each bar represents the size of the gene set. GEX: Gene expression; pop.: population. c) Box plots of adhesion ratios obtained with the washing assay for 4 of the 31 verified hits (*rpl15Δ, rpa12Δ, fkh2Δ, SPNCRNA.1234Δ*) on EMM vs YES. Each dot is an independent observation. Remaining hits are shown in Supplementary Fig. 11. d) Venn diagram showing functional conservation of the genetically conserved hits from our screen. The hits SPAC607.02c, sre2, meu27, and for3 are not shown as they are not genetically conserved (only present in *S. pombe*). Only *fkh2* and *res1* are also annotated as being involved in MLP formation in *S. cerevisiae* and *C. albicans*. Annotations are assigned using both GO and Phenotype data (GO + P). e) Scatter plot of adhesion ratios and before-wash colony intensities overlaid by a histogram showing before-wash colony intensities of nonadhesive deletion strains, which were assayed in the middle 60 spots during the original screen. The shaded area marks strains that are above the lowest 5th percentile of before-wash colony intensity in the nonadhesive strains, a proxy for growth rate. The *srb10/11Δ* strains, highlighted in green, are the most adhesive strains among those with growth values above the lowest 5th percentile. f) Line plot of growth measurements (OD600) from WT (JB22), empty media, *srb11Δ, srb10Δ, mus7Δ, rpa12Δ, kgd2Δ* and *tlg2Δ* strains in EMM and YES.

confirmed that in YES, *srb10*/11 deletions grow similarly to WT, unlike *mus7Δ* and *kgd2Δ* (Fig. 5f). In EMM, while *srb10* and *srb11* deletion mutants grow more slowly than WT strains, they grow

faster than *rpa12Δ, kgd2Δ,* and *tlg2Δ* (Fig. 5f). Thus, deleting *srb10* and *srb11* is not as detrimental to growth as deleting other genes which cause similar levels of MLP formation.

In summary, our deletion library screen for EMM agar adhesion identified 31 high-confidence hits, including genes unique to *S. pombe* as well as genes that may be functionally conserved, as they are annotated as contributing to MLP formation in budding yeasts. Additionally, the *srb11* and *srb10* null-mutants feature high levels of adhesion while maintaining better growth efficiency than these hits, possibly explaining why the *srb11* truncation can persist as a natural QTL.

# Discussion

## MLP formation as an adaptation to environmental conditions?

Although *S. pombe* has been a popular model organism for decades, its MLPs have received little attention. We find that many natural isolates exhibit MLP formation, indicating that MLPs play an important role in the natural ecology of the species. While MLPs have been understudied in *S. pombe*, there has been much more work in the distantly related budding yeasts *S. cerevisiae,* where flocculation is important for winemaking and brewing (Soares 2011), and *C. albicans*, where biofilm formation has been linked to pathogenesis (Douglas 2003; Gulati and Nobile 2016). Comparing the genes associated with MLP formation between these three species revealed several conserved proteins that regulate MLP formation, while effector cell-adhesion proteins are not conserved. The rapid evolution of cell-adhesion proteins has been noted before (Linder and Gustafsson 2008; Xie and Lipke 2010; Xie et al. 2011; Smoak et al. 2023), suggesting a possible role in adaptation to new environments and divergence of cell-cell interactions, and possibly contributing to speciation. However, it remains unclear to what extent differences in cell-adhesion proteins limit interactions between strains and even species, as different yeast species can co-flocculate (El-Behhari et al. 2000; Rossouw et al. 2015). Regardless, the contrast between highly variable effector proteins and conserved regulatory proteins is striking, given that evolution is known to rewire regulatory interactions while maintaining stable effector proteins in other pathways (Booth et al. 2010).

Although some regulatory proteins are conserved across species, their activity likely varies even within species and across different conditions. Concentrations of minerals, e.g. $Ca^{2+}$ (Matsuzawa et al. 2011), sugars (Westman et al. 2014; Ignacia et al. 2025) and pH (Su et al. 2018), can also directly affect the function of cell-adhesion proteins. Specifically, in the case of adhesion to agar, we show that different *S. pombe* strains exhibit MLP formation under different nutrient conditions, with 42% of natural isolates showing strong MLP formation in at least one condition (Fig. 2). Still, it is not clear what advantage these phenotypes confer to *S. pombe* cells, and why they vary so widely between strains. One possibility is that, similarly to *S. cerevisiae* (Koschwanez et al. 2011), aggregation in *S. pombe* allows cells to thrive in nutrient-poor environments by increasing local nutrient concentrations through a shared pool of extracellular enzymes. Indeed, we found that *mbx2* upregulation, caused by CKM deletions, results in significant upregulation of the invertase gene *inv1*, which might facilitate the sharing of digested monosaccharides as "public goods" (Koschwanez et al. 2011, 2013).

In addition, we observed a decrease in growth for MLP-forming strains both in natural isolates and the genome-wide prototrophic deletion library. This is consistent with previous work finding that two flocculating deletion strains (*rpl3201Δ* and *rpl3202Δ*) grew half as fast as wild-type *S. pombe* cells (Li et al. 2013). In that case, it is not clear if there is any causal relationship between slow growth and flocculation or if they are both parallel consequences of the ribosomal pathway disruption. Work on experimentally evolved multicellular-like *S. cerevisiae* also reported an association between group size and slow growth (Ratcliff et al. 2012). Furthermore, it has been found that *FLO1*-induced flocculating *S. cerevisiae* grows 4-times slower than its wild type counterpart (Smukalla et al. 2008), that induction of *FLO5*, *FLO9* and *FLO10* approximately halves growth rates (108), and that cAMP, a signaling molecule heavily involved in regulating growth in response to nutrient availability, also regulates filamentation (Lorenz and Heitman 1997). In this work we conclude that a tradeoff between MLP formation and growth can be detected at a global level in *S. pombe*, both in a genome-wide deletion library and in a collection of natural isolates.

The relationship between MLPs and slow growth, however, is not universal and is condition dependent. In industrial ethanol production, flocculating *S. cerevisiae* strains overexpressing *FLO5*, *FLO8* or *FLO10* show comparable growth to wild-type controls, while improving ethanol yield (Wang et al. 2023). Furthermore, in *S. cerevisiae*, MLP-forming cells grow better in low sucrose concentrations (Koschwanez et al. 2013) and under various stress conditions (Smukalla et al. 2008). Flocculating cells have also shown faster fermentation in media containing common industrial bioproduction inhibitors, despite slower fermentation than nonflocculating cells in noninhibitory media (Westman et al. 2014). We identify here another notable exception for genes in the CKM including *srb10* and *srb11*, which when deleted drive MLP formation with only a slight compromise in cell growth. In the case of *srb11* we identified a loss of function variant responsible for aggregation in two natural isolates.

## Cyclin C: genetic insight into the natural variation in MLP formation

Similar to the effects of environmental conditions, the genetic basis of natural diversity in MLP formation of *S. pombe* is also not well understood. We find that the South African strain JB759 exhibits MLP formation, which is driven by a truncation of *srb11*, encoding cyclin C, a component of the CKM of the Mediator. The canonical function of the Mediator complex is to form a bridge between general transcription factors and RNA polymerase II (Pol II), and this complex is highly conserved across eukaryotes (Tsai et al. 2013). The Mediator has four key subunits, the head, middle and tail modules forming the core Mediator, and the CKM can reversibly bind to this core (Elmlund et al. 2006; Zhu et al. 2006; Linder et al. 2008; Tsai et al. 2013). Our key finding is that in minimal media, loss of the kinase and cyclin components of the CKM, and to a lesser extent the other subunits, results in MLP formation via the upregulation of *mbx2*, encoding the transcription factor Mbx2. Mbx2 then activates expression of the flocculins as well as other genes such as the external sucrose invertase *inv1* (Supplementary Fig. 8a). These results expand on previous work that showed that deletion of *srb10* (alternatively called *prk1*) induces flocculation (Watson and Davey 1997), which was later shown to be downstream of the LAMMER kinase *lkh1* (Park et al. 2018), the deletion of which also causes flocculation (Kim et al. 2001). Recently, it was found that *mbx2* is upregulated in both of these mutants (Kang et al. 2024), and we now show the absolute requirement for Mbx2 in driving the CKM-related MLP formation, and identify a naturally occurring *srb11* truncation mutant in this pathway.

The canonical function of the CKM is to inhibit Pol II recruitment to the promoter, and thereby to repress basal transcription (Elmlund et al. 2006; Tsai et al. 2013). However, such

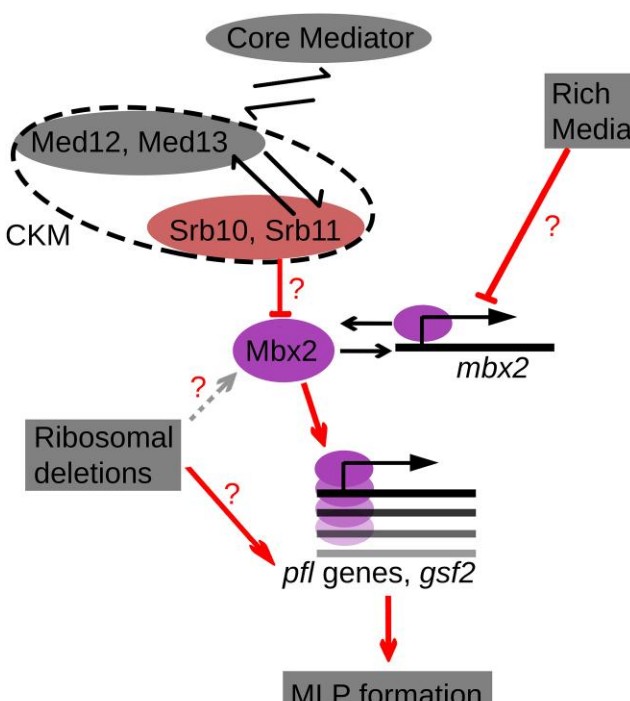

**Fig. 6.** Model for EMM-dependent MLP formation of CKM mutants. We propose that the CKM phosphorylates Mbx2 and targets it for degradation, based on similar observations in *C. albicans* (Hollomon et al. 2022) and *S. cerevisiae* (Raithatha et al. 2012). Additionally, *mbx2* expression is repressed through an unknown mechanism in YES, as we observed using RT-qPCR. In minimal media, if members of the CKM are deleted, *mbx2* becomes upregulated, triggering the expression of flocculin genes, which in turn cause MLP formation. Deletion of ribosomal genes also triggers MLP formation, although it is unclear whether this occurs through upregulation of *mbx2* or directly through the flocculin genes. The red arrows indicate our main findings, and the red question marks highlight the main outstanding mechanistic questions. See main text for details.

transcriptional repression is not thought to be gene-specific, as the DNA-binding profiles of the CKM match the broad DNA-binding profiles of the core Mediator (Zhu et al. 2006). Several instances of noncanonical functions of the CKM have been found in yeasts. In *C. albicans,* the CKM phosphorylates the hyphal growth promoting transcription factor Flo8p, which is thereby targeted for degradation, thus repressing hyphal growth and adhesion (Hollomon et al. 2022). In *S. cerevisiae,* the CKM can affect histone lysine methylation and repress the expression of the cell-surface flocculin gene *FLO11* and of the *inv1* ortholog, the sucrose invertase gene *SUC2* (Law and Ciccaglione 2015). Furthermore, in *S. cerevisiae,* the CKM phosphorylates the transcription factors Ste12p and Phd1p (Nelson et al. 2003; Raithatha et al. 2012), leading to their degradation and repression of filamentous growth. The CKM is therefore a conserved repressor of MLP formation across yeast species.

In *S. pombe*, the most studied aspect of the CKM is its regulation of mitotic entry through periodic phosphorylation of the forkhead transcription factor Fkh2 (Szilagyi et al. 2012). Interestingly, *fkh2* came up as a hit in our deletion screen, and its adhesion phenotype was also EMM-dependent, although milder than that of *srb10* and *srb11* deletions. However, deletion of *fkh2* does not appear to increase *mbx2* levels in published datasets (Supplementary Fig. 8e). Orthologs of *fkh2* are negative regulators of MLP formation in *C. albicans* and *S. cerevisiae* as well (Hollenhorst

et al. 2000; Zhu et al. 2000; Bensen et al. 2002). In *S. pombe*, the phosphorylation of Fkh2 by Srb10 inhibits its degradation, and therefore allows Fkh2 to accumulate and trigger entry into mitosis (Szilagyi et al. 2012).

Surprisingly, while a lack of Srb10 activity delays mitotic entry, deletions of *med12*/*srb8* or *med13*/*srb9* show the opposite phenotype, advancing mitotic entry (Banyai et al. 2014). The authors' explanation is that normally Med12 and Med13 anchor Srb10 and Srb11 to the Mediator, and deletion of this anchor results in an active pool of free Srb10 and Srb11, ready to phosphorylate Fkh2 (Banyai et al. 2014). Crucially, MLP formation seems to be another such phenotype that differs strikingly between deletions of different parts of the CKM, as *med13Δ* and *med12Δ* leads to milder adhesion phenotypes compared with *srb10Δ* or *srb11Δ* (Linder et al. 2008). Surprisingly, *mbx2* transcript levels seem to be upregulated in all CKM deletions for which transcriptomic data exists (*srb10Δ, srb11Δ,* and *med12Δ,* see Fig. 4e, Supplementary Fig. 8e), and it is therefore unclear why their phenotypes differ. It also remains unknown how exactly the CKM affects Mbx2. Given the noncanonical roles of the CKM in *S. cerevisiae* and *C. albicans* (Nelson et al. 2003; Raithatha et al. 2012; Hollomon et al. 2022), a possible scenario is that it phosphorylates Mbx2, which results in its degradation (Fig. 6). In this case, deletion of the CKM would result in the accumulation of Mbx2, which binds its own promoter (Kwon et al. 2012) and would therefore upregulate the *mbx2* transcript. Alternatively, the CKM might phosphorylate and stabilize a repressor of *mbx2*. Further dissection of this pathway will require phosphoproteomic data similar to that recently collected in *C. albicans* (Hollomon et al. 2022). It also remains unclear why CKM deletions result in the upregulation of *mbx2* only in minimal media, suggesting an upstream repressive mechanism in rich media (Fig. 6). Given that overexpression of *mbx2* leads to MLP-formation in rich media, this repression likely does not act at the protein level, but rather at the transcript level. Further investigation of the natural isolate JB914, which contains the *srb11* truncation while showing MLP formation in both rich and minimal media, has the potential to reveal this mechanism.

Besides the extreme adhesion phenotypes of CKM mutants, we find that deletion of genes for three other Mediator complex subunits, *med10*, *med18*, and *med19*/*rox3*, also cause mild adhesion to agar. A positive genetic interaction has previously been found between *mbx2* and *med19*/*rox3*, supporting our finding (Ryan et al. 2012). In addition to adhesion phenotypes, Mediator head mutants (including *med8, med17, med18, med20,* and *med27* deletions) display filamentous growth (Linder et al. 2008), an MLP that reflects a lack of cell separation after mitosis. This phenotype occurs through loss of expression of the transcription factor gene *ace2* (Linder et al. 2008). Interestingly, cell separation is also regulated by Ace2 in *S. cerevisiae*, and various Mediator defects cause a drop in transcription of Ace2 targets in *S. cerevisiae* (Linder et al. 2008). From all these findings, the Mediator emerges across divergent yeast lineages as a conserved central hub of MLP regulation, upstream of Mbx2 which drives expression of cell-adhesion proteins that cause separated cells to adhere to one another, and Ace2, which prevents cells from separating following division, thus driving filamentation.

Although we identified 31 genes whose deletion results in adhesion, none of these genes were present as a null mutation in natural isolate strains which exhibited MLPs, in contrast to *srb11*. We hypothesize that the benefit of *srb10* or *srb11* deletions, compared with deletions in the other genes, lies in their strong adhesion phenotypes coupled with only a slight compromise in cell growth. Therefore, if MLP formation in a low-nutrient environment is

selected for, a null mutation in *srb10* or *srb11* might be one of the most optimal outcomes given its limited growth tradeoff. Accordingly, experimental evolution in rich media (where *srb10/11Δ* does not lead to MLP formation) does not select for mutations in *srb10*/11 and shows no overlap with our results on minimal media (Pineau et al. 2024).

## Ribosomal genes and MLP formation: a novel pathway?

The 31 hits from our screen include 5 ribosomal genes (*rpl15*, *rpl3602*, *rps1201*, *rpl2102*, *rpl2702*) (Figs. 5b and 6). Li et al. (2013) also observed that deletion of three more ribosomal genes, *rpl3201*, *rpl3202*, and *rpl902*, causes MLP formation in *S. pombe*. Liu et al. (2015) then linked the *rpl3201* and *rpl3202* deletions to the upregulation of the flocculin genes (Fig. 6), which might be mediated by Mbx2 in this case as well. In our screen, *rpl3202Δ* exhibited adhesion above the 95th percentile, but its postwash intensity was below our threshold for robust quantification in the verification step, likely due to impaired growth, and it was therefore filtered out from our final hits (similar to two additional ribosomal deletions: *rpl2101Δ* and *rpl3702Δ*). Such ribosomal deletions might mimic physiological conditions of low translation (e.g. ribosomal inhibition due to toxins or starvation). Under these conditions, individual cells might not produce sufficient amounts of specific proteins that repress MLP formation. On the other hand, several of our ribosomal hits have paralogs, as many ribosomal genes in *S. pombe* do, and are therefore likely to be partly redundant. Ribosomal paralogs are tuned to certain translational responses in *S. cerevisiae* (Komili et al. 2007; Ghulam et al. 2020; Malik Ghulam et al. 2022), and in *S. pombe*, there is evidence for paralog-dependent differences in ribosome compositions (Li et al. 2022). The missing ribosomal subunits could, therefore, trigger specific metabolic signals that cause the cell to sense starvation. In the latter case, forming MLPs might be an adaptive strategy that allows starving cells to share "public goods", e.g. extracellular enzymes and metabolites. If this is a general mechanism resulting from proteome-wide decreases in translation, it is unclear why only certain ribosomal subunits triggered MLP formation in our screen, when a total of 98 ribosomal gene deletions were assayed (as captured by GO:0005840 ribosome). As this phenomenon has not been reported in *S. cerevisiae* or *C. albicans,* MLP formation caused by the deletion of these genes might be unique to *S. pombe*.

## Conclusion

In this work, we generated valuable datasets that will serve as the basis for future mechanistic studies of MLP formation in *S. pombe*. Additionally, our work makes the first step towards understanding the natural diversity of MLP formation in fission yeast. Furthermore, we report novel players in MLP formation, some of which might represent pathways unique to *S. pombe*, and others which are conserved in other yeasts. Finally, our high-throughput flocculation and surface-adhesion assays are applicable to other microbes, and due to their high-throughput nature, they could be used to uncover the diversity in MLP formation both within and across species.

## Data availability

All collected data, performed analyses, and the sequences of the primers used have been deposited to both Github (https://github.com/BKover99/S.-Pombe-MLPs) and Figshare (https://doi.org/10.6084/m9.figshare.25750980). Most analyzes are available

in a Jupyter notebook format (.ipynb). QTL analysis is available as an R script, while the haplotype calling pipeline is available as a bash script. The analysis tools used for our high-throughput assays can be accessed as a standalone package from https://github.com/BKover99/yeastmlp and can be installed from PyPI using the command "pip install yeastmlp".

Supplemental material available at GENETICS online.

## Acknowledgments

We thank M. Lera-Ramirez and M. D'Angiolo for critical comments on the manuscript and S. Marguerat, D. Ellis, A. Garg, D. Jeffares, O. Hillson, B. Schwer, and S. Shuman for insightful discussions and sharing protocols and unpublished results. We thank the University College London Center for Cell and Molecular Dynamics Core Facility (RRID:SCR_026564) for training and access to the Zeiss Cell Discoverer 7 microscope.

## Funding

This research was supported by a Wellcome Discovery Award (302608/Z/23/Z) to J.B. C.E.C. was supported by a Genetics Society Summer Studentship. B.M.H. and M.R. were supported by Wellcome Trust grant (IA 200829/Z/16/Z).

## Conflicts of interest

None declared.

## Author contributions

B.M.H., B.K., and J.B. conceptualized the work. B.M.H., B.K., and C.E.C. developed the methodology. B.K., L.S., C.E.C., S.R. and B.M.H. conducted experiments. B.K., B.M.H., C.E.C., and L.S. performed analysis. B.K. and B.M.H. wrote the original draft and prepared figures. B.K., B.M.H., C.E.C., and J.B. revised the manuscript. J.B. and B.M.H. provided supervision and J.B. and M.R. provided funding.

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

*Editor: P. Wittkopp*