## [Peer Review File · Genetics]

Genetic and environmental determinants of multicellular-like phenotypes in fission yeast

Bence Kövér, Céleste Cohen, Ladislav Seres, Saniya Raut, Markus Ralser, Benjamin Heineike, and Jürg Bähler

NOTE: The reviews and decision letters are unedited and appear as submitted by the reviewers.

In extremely rare instances and as determined by a Senior Editor or the EIC, portions of a review may be redacted. If a review is signed, the reviewer has agreed to no longer remain anonymous.

The review history appears in chronological order.

Review Timeline:

Submission Date:

2026-01-28

Accepted:

2026-02-19

EDITORIAL NOTE: This manuscript and reviews were transferred to GENETICS from Review Commons. The anonymized reviews and the authors' response to reviewer comments are included in this document.

Review
COMMONS

Review #1

1. Evidence, reproducibility and clarity:

Evidence, reproducibility and clarity (Required)

****Summary****

Köver et al. examine the genetic and environmental underpinnings of multicellular-like phenotypes (MLPs) in fission yeast, studying 57 natural isolates of *Schizosaccharomyces pombe*. They uncover that a noteworthy subset of these isolates can develop MLPs, with the extent of these phenotypes varying according to growth media. Among these, two strains demonstrate pronounced MLP across a range of conditions. By genetically manipulating one strain with an MLP phenotype (distinct from the previously mentioned two strains), they provide evidence that genes such as MBX2 and SRB11 play a direct role in MLP formation, strengthening their genetic mapping findings. The study also reveals that while some key genes and their phenotypic effects are strikingly similar between budding and fission yeast, other aspects of MLP formation are not conserved, which is an intriguing finding.

Overall, the manuscript is well-written, dense yet logically structured, and the figures are well presented. The combination of phenotypic, genetic, and bioinformatics analyses, particularly from wet lab experiments, is commendable. The study addresses a significant gap in our understanding, primarily explored in budding yeast, by providing comprehensive data on MLP diversity in fission yeast and the interplay of genetic and environmental factors.

In summary, I enjoyed reading the manuscript and have only a few minor suggestions to strengthen the paper:

****Minor revisions:****

1. Although this may seem like a minor revision, but it is a crucial point. Please make sure that all raw data used to generate figures, run stats, sequence data, and scripts used to run data analysis are made publicly available. Provide relevant accession numbers and links to public data repositories. It is important that others can download the various types of data that went into the major conclusions of this paper in order to replicate your analysis or expand upon the scope of this work. I am not sure if the journal has a policy regarding this, but it should be followed to allow for transparency and reproducibility of the research.
2. Two out of 57 strains exhibit strong and consistent MLP across multiple environments. Providing more information on these strains (JB914 and JB953), such as their natural habitats and distinct appearances of their MLP phenotypes under varying conditions, would provide valuable insights.

First, a brief discussion highlighting what differentiates these two strains from the rest would be helpful for readers (e.g. insight into their unique genetic and environmental background that might be linked to the MLP phenotype).

Additionally, culture tube and microscopy images of these strains, similar to those presented for JB759 in Figure 2A, can be included in the supplementary materials. My reasoning is that these images could help illustrate variation or lack thereof in aggregative group size across different media.

3. The phenotypic outcome of overexpressing MXB2 is striking, as shown in Supplementary Figure 4C. Incorporating at least one of the culture tube images depicting large flocs into the main text, perhaps adjacent to Figure 3 panel D, would improve the visual appeal and highlight this key finding (at the moment those images are only shown in the supplementary materials).
4. I know that the authors discuss the knowledge gap in the intro and results, but the abstract does not mention this critical gap. Please stress this critical gap (i.e., MLPs understudied in fission yeast) with a brief sentence in the abstract. Similarly, please consider writing a brief concluding sentence summarizing the paper's most significant finding referring to the knowledge gap would provide a clearer takeaway message for the reader - the abstract ends abruptly without any conclusion.
5. The observation that strains with adhesive phenotypes have a lower growth rate compared to non-adhesive strains is a noteworthy point (lines 532-535). This represents yet another example of this classical trade-off. This point could be emphasized in the Discussion or alongside the relevant result, with a brief speculative explanation for this phenomenon.
6. The text mentions two lab strains, JB22 and JB50, displaying strong adhesion under phosphate starvation (lines 525-526), yet the data point for JB22 in Figure 2C is not labeled.
7. Although I generally avoid commenting on formatting, I found the manuscript to be dense. As mentioned above, I truly enjoyed reading it! But I couldn't help but think of ways to make the manuscript more concise for readers. The Results section spans nine pages (excluding figure captions), and the Discussion is five pages long. The main text contains 6 figures with approximately 27 panels and 32 plots and Venn diagrams, while the supplementary material has 11 figures with 22 panels and about 59 plots. Altogether, the manuscript comprises 17 figures, 49 panels, and roughly 91 plots and Venn diagrams! While I will not request any changes, I encourage the authors to consider streamlining the text/data where possible to focus on the core theme of the study.

****Referees cross-commenting****

There are many useful recommendations from all the other reviewers that will help improve the final product. Once those points are revised, I think this will be a nice paper of interest to folks interested in natural variation in MLPs and its genetic background.

2. Significance:

Significance (Required)

My expertise: evolutionary genetics, evolution of multicellularity, yeast genetics, experimental evolution

Overall, the manuscript is well-written, dense yet logically structured, and the figures are well presented. The combination of phenotypic, genetic, and bioinformatics analyses, particularly from wet lab experiments, is commendable. The study addresses a significant gap in our understanding, primarily explored in budding yeast, by providing comprehensive data on MLP diversity in fission yeast and the interplay of genetic and environmental factors.

In summary, I enjoyed reading the manuscript and have only a few minor suggestions to strengthen the paper.

3. How much time do you estimate the authors will need to complete the suggested revisions:

Estimated time to Complete Revisions (Required)

(Decision Recommendation)

Review #2

1. Evidence, reproducibility and clarity:

Evidence, reproducibility and clarity (Required)

Yeast species, including fission yeast and budding yeast, could form multicellular-like phenotypes (MLP). In this work, Kövér and colleagues found most proteins involved in MLP formation are not functionally conserved between *S. pombe* and budding yeast by bioinformatic analysis. The authors analyzed 57 natural *S. pombe* isolates and found MLP formation to widely vary across different nutrient and drug conditions. The authors demonstrate that MLP formation correlated with expression levels of the transcription factor gene *mbx2* and several flocculins. The authors also show that *Cdk8* kinase module and *srb11* deletions also resulted in MLP formation. The experimental design is logic, the manuscript is well-written and organized. I have a few concerns that should be addressed before the publication.

Major points:

1. Line 61-62, how did the authors grow yeast cells in the liquid medium? Shaking or static? If shaking, the nutrient should be even distributed in the medium.
If static culture, most single yeast cells could precipitate on the bottom, how do you address the advantage of flocculation for increasing the sedimentation? In addition, under static culture, the bottom will have less air than the up medium, how to balance the air and nutrients?
2. Line 555, it will be interesting to test whether overexpression of *mbx2* could cause flocculation in YES medium. In Figure 3D, the authors use two control strains, but only one *mbx2* OE strain, *mbx2* OE should be tested in both strains. In addition, did the authors transform empty plasmid into the control strains, please indicate in the figure.
3. Line 600-601, the authors may do the backcross of *srb11*Δ::Kan to exclude the possibility caused by other mutations.

Minor points:

1. Line 506, what are the growth conditions of cells in Figure 2A? Did the authors use the liquid or solid medium? Please mention in the Methods or figure legends.
2. Line 533-535, please explain why the strains exhibiting strong adhesion have a decreased growth rate. Is there

any related research? Please add some references.

****Referees cross-commenting****

I agree with most of the comments from other reviewers. This publication may indeed be of interest to a minor area. But the results and the interpretations of the data are interesting and warranted, the findings are scientifically important.

2. Significance:

Significance (Required)

The authors did many large-scale screens and bioinformatic analyses. The experiments in the manuscript are generally logical and sound. This study is useful for deciphering the mechanism of multicellular-like phenotype formation in the fission yeast, with some implications for some other organisms.

3. How much time do you estimate the authors will need to complete the suggested revisions:

Estimated time to Complete Revisions (Required)

(Decision Recommendation)

Between 1 and 3 months

Review #3

1. Evidence, reproducibility and clarity:

Evidence, reproducibility and clarity (Required)

****Summary:****

Using a variety of targeted and genome wide analyses, the authors investigate the basis for "multicellular-like phenotypes" in *S. pombe*. Authors developed several methodologies to detect and quantify "multicellular-like phenotypes" (flocculation, aggregation...) and defined genes involved in these processes in laboratory and wild *S. pombe*.

SECTION A - Evidence, reproducibility and clarity

This is a very solid manuscript that is well-written and supported by convincing data. While one can imagine many additional experiments, the manuscript stands on its own and presents a quite exhaustive analysis of the area. I commend the author for their rigorous work and clear presentation. They are only a few minor points that warrant comments or corrections:

- Supplementary Figure 1 is a typical example of the "necessity" to have statistics and P-values everywhere. The data are convincing but what is the evidence that the Filtering assay and the Plate-reader assay values should be linearly related? Lets imagine that Plate-reader assay value is proportional to the square of the Filtering assay value. What would be the Pearson R and P-value in this case? What is most appropriate? Why would one use a linear correlation? What is the "real" significance?

****Minor points:****

- They are several "personal communications" in the manuscript (page 11, page 18, page 23). It should be checked whether this is accepted in the journal that publishes this manuscript.
- Page 4 check "a few regulators"
- Page 19, line 567: "remaining 8 strains" may be confusing as Material and Methods states "remaining 10 strains".

****Referees cross-commenting****

I concur with most comments. Overall, the reviewers agree that this is a solid piece of work that could benefit from minor modifications and should be published. I reiterate that, for me, despite its quality, this publication will only be of interest to specialists.

2. Significance:

Significance (Required)

A limited number of studies have investigated "multicellular-like phenotypes" in *S. pombe*. This manuscript brings therefore new and solid information. Yet, despite an impressive amount of work, our conceptual advance in understanding this process and its phylogenetic conservation remains limited. This is probably best illustrated in the figure 6 that summarize the study and contains 3 question marks and an additional unknown mechanism. (Most of the solid arrows in this figure correspond to interactions within the Mediator complex that were well known before this study.) In addition, while only few studies have been published in this area, the authors' findings are often only bringing additional support to already published observations.

Overall, while this manuscript will be of interest to a restricted group of aficionados, it will most likely not attract the attention of a wide readership.

3. How much time do you estimate the authors will need to complete the suggested revisions:

Estimated time to Complete Revisions (Required)

(Decision Recommendation)

Less than 1 month

Review #4

1. Evidence, reproducibility and clarity:

Evidence, reproducibility and clarity (Required)

In this manuscript, the authors explore how multicellular-like phenotypes (MLPs) arise in the fission yeast *S. pombe*. Although yeasts are characterized as unicellular fungi, diverse species show MLPs, including filamentous growth on agar plates and flocculation in liquid media. MLPs may provide certain advantages in nutritionally poor conditions and protection against external challenges, upon which natural selection can then act. Previous work on MLPs has mostly been carried out in the budding yeasts *S. cerevisiae* and *C. albicans*, and little was known about these behaviors in *S. pombe*. The authors thus set out to investigate both genetic and environmental regulators of MLP formation.

First, their analysis of published data revealed a limited number of shared regulators of MLP between *S. pombe*, *S. cerevisiae*, and *C. albicans*, although the cell adhesion proteins themselves are largely not conserved. Next, the authors screened a set of non-clonal natural isolates using two high-throughput assays that they developed and found that MLPs vary in strains and depending on nutrient conditions. Focusing on a natural isolate that showed both adhesion on agar plates and flocculation in liquid medium, they then analyzed a segregant library generated from this and a laboratory strain using their assays. Using QTL analysis, they uncovered a frameshift in the *srb11* gene, which encodes a subunit of the Mediator complex, as the likely causal inducer of MLP. This was confirmed by additional analyses of strains lacking *srb11* or other members of Mediator. Furthermore, the authors showed that loss of *srb11* function resulted in the upregulation of the Mbx2 transcription factor, which was both necessary and sufficient for MLP formation in this background. Finally, screening of two additional yeast strain collections (gene and long intergenic non-coding RNA deletion) identified both known and novel regulators representing different pathways that may be involved in MLP formation.

Altogether, this study provides new perspectives into our understanding of the diverse inputs that regulate multicellular-like phenotypes in yeast.

****Major comments:****

- The methods for screening for adhesion and flocculation are well described, with representative figures that show plates and flasks. However, there are few microscopy images of cells, and it would be interesting and helpful for the reader to have an idea of how cells look when they exhibit MLPs. For instance, are there any differences in cell shape or size when strains present different degrees of adhesion or flocculation? In addition, the authors mention that mutants with strong adhesion generally had lower colony density and are likely to be slower growing. Although their analyses suggest otherwise (page 22), this has a potential for introducing error in their observations, and including images of the adhesion/flocculation phenotypes may provide further support for their conclusions. I suggest that the authors present microscopy images 1) similar to what is shown for JB759 in Figure 2A and 2) of cells growing on agar in the adhesion assay. This could be included for the different Mediator subunit deletions that they tested, where there appear to be varying phenotypes. It could also be informative for a subset of the 31 high-confidence candidates that they identified in their screen.
- Upon identifying a frameshift in *srb11* that is responsible for the MLP, the authors assessed whether deletion of other Mediator subunits would result in the same phenotype. They found that *srb10* and *srb11* deletions both flocculate and show adhesion, while other mutants had milder phenotypes. However, the authors also found that a new deletion of *srb11* that they generated had a stronger adhesion phenotype than the *srb11* deletion from the prototrophic deletion library, which was attributed this the accumulation of suppressor mutations in the strains of the deletion collection. As the authors make clear distinctions between the phenotypes of different Mediator mutants, I suggest generating and analyzing "clean" deletions of the 6 other subunits that they tested. This would strengthen their conclusion and help to rule out accumulated suppressors as the cause of the differences in the observed phenotypes.

****Minor comments:****

- One point that recurs in the manuscript is the idea that mutations that give rise to strong MLPs also generally lead to slower growth, representing a potential trade-off. This idea could be reinforced with measurements of growth rate or generation time by optical density or cell number, for instance, rather than comparisons of colony density. Also, it would be interesting to mention if the slow growth phenotype is only observed in MLP-inducing conditions or also in rich medium.
- The authors show that the MLPs of the *srb10* and *srb11* deletions occur through *mbx2* upregulation. Do the varying strengths of the phenotypes of the strains lacking different Mediator subunits correlate with *mbx2* levels in these backgrounds?

****Referees cross-commenting****

I agree overall with the comments and suggestions from the other reviewers. The revision would require only minor modifications. The paper is interesting both for the combination of methodologies used and its findings, and I believe that it would benefit a growing community of researchers.

2. Significance:

Significance (Required)

This study employed a variety of methods that allowed the authors to uncover previously unknown regulators of MLPs. Taking advantage of the diversity of natural fission yeast isolates as well as the constructed gene and non-coding RNA deletion collections, the authors identified novel genetic determinants that give rise to MLPs, opening new avenues into this exciting area of research. The overall conclusions of the work are solid and supported by the reported results and analyses. This study will be appreciated by a broad audience of readers who are interested in understanding how organisms respond to environmental challenges as well as how MLPs

may result in emergent properties that play key roles in these responses. Some of the limitations of the work are described above, with recommendations for addressing these points.

Keywords for my field of expertise: fission yeast, cell cycle, transcription, replication.

3. How much time do you estimate the authors will need to complete the suggested revisions:

Estimated time to Complete Revisions (Required)

(Decision Recommendation)

Between 1 and 3 months

Full Revision

Manuscript number: RC-2023-02318

Corresponding author(s): Benjamin Heineike, Jürg Bähler

1. General Statements [optional]

We thank the reviewers and editor for their time taken in reviewing our manuscript “Genetic and environmental determinants of multicellular-like phenotypes in fission yeast” and for their helpful, constructive comments. We have thoroughly addressed the issues raised by further experiments and text changes, which have improved the manuscript. Our detailed responses to the reviewers’ suggestions are given below in blue.

We acknowledge that it has taken a long time to complete these revisions. This was due to several factors. The first author, Bence Kövér, had moved to another research group, and Ben Heineike had primary employment in a different lab until August of 2025, and had to focus on another project. Moreover, our lab moved to a new building during the revisions. Finally, we prioritised completing some of the experiments and analysis suggested by reviewers over the speed of response. The most challenging of these was rebuilding the *mbx2* overexpression strains that we could use in rich media to answer Reviewer 2’s important question about whether the phenotype of *mbx2* overexpression could be repressed in rich media. Our new experiments emphatically show that that is not the case, and we believe this provides important additional context for our model. We have added new authors, Ladislav Seres and Saniya Raut, who helped with the experiments.

Despite the delay, the resulting manuscript has not lost its novelty and indeed has been refined through the process into an even more striking contribution to the field. We are heartened by the fact that the manuscript has had over 2200 abstract views and >1400 downloads from biorxiv, and we believe that publication in a journal, along with the improvements we have made in this process, will increase its impact and visibility.

The ability of single-celled organisms to adopt Multicellular-Like Phenotypes (MLPs), from flocculation, to adhesion and biofilm formation, has consequences in understanding and manipulating them in the context of human health and bioproduction. The fission yeast, *S. pombe*, is a popular single-celled eukaryotic model organism, but it has not been as closely studied in the context of MLP formation as its distant relatives, the budding yeasts *S. cerevisiae* and *C. albicans*. In this manuscript, we provide insights into MLP formation in *S. pombe*. We show how a specific naturally occurring frameshift mutation in the *srb11* gene, a cyclin in the Cdk8 Kinase Module of the Mediator, can drive MLP formation without experiencing a severe decrease in growth rate. We also identify 31 gene deletions that cause cell adhesion, including 15 that had not previously been associated with cell adhesion in any organism. In addition to presenting new findings, our study integrates and synthesizes previously fragmented work on multicellular-like phenotypes in *S. pombe*, providing a comprehensive survey of the field. As such, it serves as a unifying reference point for future research on this topic.

Thank you for your assistance and patience.

Best regards,

Jürg Bähler

Ben Heineike

Bence Kövér

Full Revision

This section is mandatory. Please insert a point-by-point reply describing the revisions that were already carried out and included in the transferred manuscript.

Summary

Köver et al. examine the genetic and environmental underpinnings of multicellular-like phenotypes (MLPs) in fission yeast, studying 57 natural isolates of *Schizosaccharomyces pombe*. They uncover that a noteworthy subset of these isolates can develop MLPs, with the extent of these phenotypes varying according to growth media. Among these, two strains demonstrate pronounced MLP across a range of conditions. By genetically manipulating one strain with an MLP phenotype (distinct from the previously mentioned two strains), they provide evidence that genes such as MBX2 and SRB11 play a direct role in MLP formation, strengthening their genetic mapping findings. The study also reveals that while some key genes and their phenotypic effects are strikingly similar between budding and fission yeast, other aspects of MLP formation are not conserved, which is an intriguing finding.

Overall, the manuscript is well-written, dense yet logically structured, and the figures are well presented. The combination of phenotypic, genetic, and bioinformatics analyses, particularly from wet lab experiments, is commendable. The study addresses a significant gap in our understanding, primarily explored in budding yeast, by providing comprehensive data on MLP diversity in fission yeast and the interplay of genetic and environmental factors.

In summary, I enjoyed reading the manuscript and have only a few minor suggestions to strengthen the paper:

Minor revisions:

1. Although this may seem like a minor revision, but it is a crucial point. Please make sure that all raw data used to generate figures, run stats, sequence data, and scripts used to run data analysis are made publicly available. Provide relevant accession numbers and links to public data repositories. It is important that others can download the various types of data that went into the major conclusions of this paper in order to replicate your analysis or expand upon the scope of this work. I am not sure if the journal has a policy regarding this, but it should be followed to allow for transparency and reproducibility of the research.

Reply: We very much agree with the reviewer that sharing raw data and scripts is an essential part of open science. All code and data are deposited to Github (<https://github.com/BKover99/S.-Pombe-MLPs>) and Figshare (https://figshare.com/articles/software/S_-Pombe-MLPs/25750980), which have now been updated to reflect our revisions. Additionally, the sequenced genomes have been deposited to ENA (PRJEB69522). Where external data was used, it was properly referenced and specifically included in Supplementary Table 3.

2. Two out of 57 strains exhibit strong and consistent MLP across multiple environments. Providing more information on these strains (JB914 and JB953), such as their natural habitats and distinct appearances of their MLP phenotypes under varying conditions, would provide valuable insights.

First, a brief discussion highlighting what differentiates these two strains from the rest would be helpful for readers (e.g. insight into their unique genetic and environmental background that might be linked to the MLP phenotype).

Additionally, culture tube and microscopy images of these strains, similar to those presented for JB759 in Figure 2A, can be included in the supplementary materials. My reasoning is that these images could help illustrate variation or lack thereof in aggregative group size across different media.

Reply: We thank the reviewer for highlighting this issue. Our further investigation into these strains has added additional interesting insights. JB914 and JB953 were isolated from molasses in Jamaica and the exudate of Eucalyptus in Australia, respectively, though it remains unclear whether these environments are related or even selective for the ability of these strains to form MLPs. We note that the environment from which a strain is isolated is an incomplete way of assessing its ecology. Indeed, recent research suggests that the primary habitat of *S. pombe* is honeybee honey and suggests that bees, which may be attracted to a number of sugary substances, may be a vector by which fission yeast are transported (1). Therefore, isolation from a particular nectar or food production environment might not reflect significant ecological differences. We now refer to the location of strain isolation in the manuscript text (lines 208-209).

However, there is more to learn from the genetic backgrounds of these two strains. We found that JB914 possesses the same variant in *srb11* causally related to MLPs as JB759, the MLP-forming parental strain for our QTL analysis. To understand whether the appearance of this variant in these two strains derived from a single mutation event or was a case of convergent evolution, we analysed homology between the genomes of JB759 and JB914, focusing specifically on that variant. We found an approximately 20kb region of homology between JB759 and JB914 surrounding the *srb11* truncation variant, in contrast to the majority of the genome, which does not share homology between those two strains (New Supplementary Figure 9A, B)). This result suggests that, while the two strains are largely unrelated, that specific region shares a recent common ancestor and is likely a result of interbreeding across strains.

Importantly, this analysis further emphasizes the point that the *srb11* variant segregates with the MLP-forming phenotype. We conclude this because none of the other strains similar to JB759 (either across the whole genome, or specifically in the region surrounding *srb11*) exhibit MLPs (New Supplementary Figure 9C). This thereby further complements our QTL analysis on the significance of this variant. We have added this analysis to the manuscript text (lines 337-349).

Furthermore, we searched other strains which exhibited MLPs in our experiments (e.g. JB953) for frame shifts, insertions or deletions in any other genes in the CKM module or in the genes that were identified in our deletion library screen as adhesive, and did not identify any severe mutations falling into coding regions (other than the *srb11* truncation in JB914 and JB759). This indicates that MLPs in these other strains may be caused by differences in regulatory regions surrounding these genes, or variants in other genes that were not identified in our screen. We have added this analysis to our manuscript (lines 424-425) and Supplementary Table 13.

We agree that microscopy and culture tube images of JB914 and JB953 may give insight into the nature of the MLPs exhibited by those strains. We have included such images of cultures grown in YES, EMM and EMM-Phosphate media in our revision (Lines 207-208, Supplementary Figures 4 and 5). These images are consistent with our adhesion assay screen and show that JB914 and JB953 are adhesive at the microscopic level in the relevant conditions (EMM or EMM-Phosphate).

A**B****C**
New Supplementary figure 9: The *srb11* variant is present in the JB759 and JB914 natural isolates, and segregates with MLP-formation across natural isolates closely related to those strains.

(A) *S. pombe* reference genome visualised for each of the three chromosomes, with vertical lines marking variant loci (called in (2)) compared to the reference genome in at least one of the two strains. Variant loci with the same alternative allele for JB759 and JB914 are marked with purple, while variant loci where alleles do not match are marked with orange. (B) Inset showing the region surrounding the causal *srb11* variant. The +/- 20Kb window is highlighted. Vertical lines indicate variant alleles as in A. (C) Dendrogram illustrating genomic similarity of natural isolates based on the +/- 20Kb window around the *srb11* frameshift mutation. Red text indicates strains closely related to JB759 considering the whole genome (based on (2)), while blue text indicates strains related to JB914. Heatmap below indicates adhesion ratio in EMM.

3. The phenotypic outcome of overexpressing MXB2 is striking, as shown in Supplementary Figure 4C. Incorporating at least one of the culture tube images depicting large flocs into the main text, perhaps adjacent to Figure 3 panel D, would improve the visual appeal and highlight this key finding (at the moment those images are only shown in the supplementary materials).

Reply: We thank the reviewer for this suggestion. In response to Reviewer 2's suggestion to overexpress *mbx2* in YES, we created new *mbx2* overexpression strains that could overexpress *mbx2* in YES, which was not possible in our previous strain in which *mbx2* overexpression was triggered by removal of thymine from the media. We have replaced our original data from Figure 3D with data from the new *mbx2* overexpression experiment, including flask images.

4. I know that the authors discuss the knowledge gap in the intro and results, but the abstract does not mention this critical gap. Please stress this critical gap (i.e., MLPs understudied in fission yeast) with a brief sentence in the abstract. Similarly, please consider writing a brief concluding sentence summarizing the paper's most significant finding referring to the knowledge gap would provide a clearer takeaway message for the reader - the abstract ends abruptly without any conclusion.

Reply: We agree and have now emphasized the critical gap in our abstract:

"As MLP formation remains understudied in fission yeast compared to budding yeast, we aimed to narrow this gap." at lines 18-19.

Additionally, we added the following final sentence to give the reader a clearer takeaway message:

"Our findings provide a comprehensive genetic survey of MLP formation in fission yeast, and a functional description of a causal mutation that drives MLP formation in nature." at lines 31-32.

5. The observation that strains with adhesive phenotypes have a lower growth rate compared to non-adhesive strains is a noteworthy point (lines 532-535). This represents yet another example of this classical trade-off. This point could be emphasized in the Discussion or alongside the relevant result, with a brief speculative explanation for this phenomenon.

Reply: We agree that the nature of the trade-off between MLP formation is an interesting discussion point that could arise from our work. Understanding this trade-off is made more complicated by the fact that growth is always condition-dependent, and measuring growth in

strains exhibiting MLPs is non-trivial, as adhesion to labware and thick clumps of cells separated by regions of cell-free media can add variability. Nonetheless, there has been some previous work on this problem. In *S. cerevisiae*, it was shown that larger group size correlates with slower growth rate (3), and that flocculating cells grow more slowly (4). In *S. cerevisiae*, cAMP, a signalling molecule heavily involved in regulating growth in response to nutrient availability, also regulates filamentation (5). However, the relationship between flocculation and slow growth is not consistent in the literature. In some settings overexpressing the flocculins *FLO8*, *FLO5*, and *FLO10* results in slower growth (6), while in others it does not (7). In addition, ethanol production has been shown to improve for biofilms (7).

Furthermore, in *S. cerevisiae*, MLP-forming cells grow better in low sucrose concentrations (8) and under various stress conditions (4). Flocculating cells have also shown faster fermentation in media containing common industrial bioproduction inhibitors, despite slower fermentation than non-flocculating cells in non-inhibitory media (9). However, any consequence of this possible advantage on growth has not been characterised.

In *S. pombe*, there is less work on this topic; however, it has been shown that deletions of *rpl3201* and *rpl3202*, which code for ribosomal proteins, cause flocculation and slow growth (10). In that case, it is not clear if there is any causal relationship between slow growth and flocculation or if they are both parallel consequences of the ribosomal pathway disruption. We have added some of these points to the portion of the discussion that discusses this tradeoff (Lines 477-499).

To get a better understanding of this tradeoff in our system, we took several approaches. First, we added a supporting analysis (New Supplementary Figure 12B), using published growth data based on measurements on agar plates for the *S. pombe* gene deletion library (11). There, the authors defined a set of deletion strains that grow more slowly on EMM than the wild-type lab strain. We found that our MLP hit strains were significantly enriched in this “EMM-slow” category. This information is now included in the manuscript (Lines 409-413, New Supplementary Figure 12B).

New Supplementary Figure 12B: Histogram of mean fitness values, based on growth in solid EMM media, for each strain as defined in (11). MLP-forming hit strains are shown in purple, and all other strains are in yellow. Blue and red dotted lines show the thresholds beyond which strains were defined as slow- or fast-growing, respectively.

It is, however, possible that for the assays from that work, the appearance of slow growth on solid agar in adhesive cells could be partially artifactual. Indeed, we have observed that adhesive cells tend to stick to flasks and, when grown on agar plates, cells in the same colony can stick to one another rather than to inoculation loops or pin pads. Both of these dynamics can reduce initial inoculation densities. This is less of a concern for our adhesion assay and Figures 2E, 5B, and 5F, because our before-wash intensity was done with a 7x7 pinned square about 10x10 mm². Nonetheless, as we wanted to make a point about *srb10* and *srb11* mutants growing faster than other deletion mutants that exhibit MLP-formation, we also conducted growth assays in liquid media (New Figure 5F).

We observed that *srb10* Δ and *srb11* Δ strains (which exhibit MLPs in EMM) show growth curves similar to wild-type cells in minimal (EMM) and rich media (YES). On the other hand, other strains that grow similarly to wild type cells in YES, such as *tlg2* Δ and *rpa12* Δ , grow much more slowly in EMM when they clump together. There are also some strains, *mus7* Δ and *kgd2* Δ , that grow more slowly in both YES and EMM but are only adhesive in EMM.

Figure 5F: Line plot of growth measurements (OD600) from WT (JB22), empty media, *srb11* Δ , *srb10* Δ , *mus7* Δ , *rpa12* Δ , *kgd2* Δ and *tlg2* Δ strains in EMM and YES.

6. The text mentions two lab strains, JB22 and JB50, displaying strong adhesion under phosphate starvation (lines 525-526), yet the data point for JB22 in Figure 2C is not labeled.

Reply: We agree that highlighting JB22 on the figure is crucial, given that it was mentioned in the main text. JB22 is now highlighted in green on Fig 2C.

7. Although I generally avoid commenting on formatting, I found the manuscript to be dense. As mentioned above, I truly enjoyed reading it! But I couldn't help but think of ways to make the manuscript more concise for readers. The Results section spans nine pages (excluding figure captions), and the Discussion is five pages long. The main text contains 6 figures with approximately 27 panels and 32 plots and Venn diagrams, while the supplementary material has 11 figures with 22 panels and about 59 plots. Altogether, the manuscript comprises 17 figures, 49 panels, and roughly 91 plots and Venn diagrams! While I will not request any changes, I encourage the authors to consider streamlining the text/data where possible to focus on the core theme of the study.

We thank the reviewer for these suggestions and have reorganised some of our figures and text to appear less dense. We have also added several figures and panels in response to reviewer comments. While we endeavor to make our points clear and concise in the main figures, we believe that it is important to retain key supplementary figures so that an interested reader can evaluate the data in more detail:

A summary of our major changes to the figures is below, and we also provide a manuscript with changes tracked for the reviewers' convenience:

Fig 2:

- Added Panel E in response to reviewer comments.

Fig 3:

- Removed axes for *pfl3* and *pfl7* from Fig 3C, as the point was made by the other genes displayed (*mbx2*, *pfl8* and *gsf2*)
- Replaced Fig 3D with similar data from an improved experiment in response to reviewer comments.
- Added New Fig 3F from Original Supp Fig 5

Fig 5:

- Moved Original Fig 5A to New Supp Fig 10A.
- Added New Fig 5F in response to reviewer comments.

Original Supp Fig 4 / New Supp Fig 6:

- Removed *mbx2* overexpression images from Original Fig 4C, to be replaced by new overexpression data and images in New Fig 3D.
- Added flask images for *srb10* and *srb11* deletion mutants from Original Supp Fig 5A to New Supp Fig 6C.
- Added microscope image for *srb11* deletion mutant from Original Supp Fig 5A to New Supp Fig 6C.
- Added adhesion assay results from Original Supp Fig 5C to New Supp Fig 6C.
- Added New Supp Fig 6D in response to review

Original Supp Fig 5

- Removed this figure. Original Supp Fig 5A and 5B were moved to New Supp Fig 6. Original Supp Fig 5B was removed to make the manuscript more concise.

Original Supp Figs 6, 7 and 8 were combined into New Supp Fig 8.

- Original Supp Fig 6A and 6B are now New Supp Fig 8A and 8B.
- Original Supp Fig 7 is now New Supp Fig 8C.

- Original Supp Fig 8A is now New Supp Fig 8D and 8E.
- Original Supp Fig 8B is now New Supp Fig 8F

Original Supp Fig 9/New Supp Fig 10

- Added Original Fig 5A as new Supp Fig 10A.

Original Supp Fig 11/New Supp Fig 12

- Removed Original Fig 11B and the relevant text to make the manuscript more concise.
- Added New Supp Fig 12B in response to reviewer comments.

New Supplementary Figures added in response to reviewer comments:

- New Supp Fig 4: Microscopy images of natural isolates.
- New Supp Fig 5: Flask images of natural isolates
- New Supp Fig 7: Microscopy and flask images of *mbx2* overexpression strains.
- New Supp Fig 9: Genomic comparisons between JB759 and the MLP-forming wild isolate, JB914.

Removed some less relevant points from our discussion, to reduce the length.

Added new Supplementary Tables:

- Supplementary Table 13: Variants in candidate genes. Added in response to reviewer comments
- Supplementary Table 14: List of plasmids used in the study.

****Referees cross-commenting****

There are many useful recommendations from all the other reviewers that will help improve the final product. Once those points are revised, I think this will be a nice paper of interest to folks interested in natural variation in MLPs and its genetic background.

Significance

My expertise: evolutionary genetics, evolution of multicellularity, yeast genetics, experimental evolution

Overall, the manuscript is well-written, dense yet logically structured, and the figures are well presented. The combination of phenotypic, genetic, and bioinformatics analyses, particularly from wet lab experiments, is commendable. The study addresses a significant gap in our understanding, primarily explored in budding yeast, by providing comprehensive data on MLP diversity in fission yeast and the interplay of genetic and environmental factors.

In summary, I enjoyed reading the manuscript and have only a few minor suggestions to strengthen the paper.

Reviewer #2

Evidence, reproducibility and clarity

REVIEWER COMMENTS

Yeast species, including fission yeast and budding yeast, could form multicellular-like phenotypes (MLP). In this work, Kövér and colleagues found most proteins involved in MLP formation are not functionally conserved between *S. pombe* and budding yeast by bioinformatic analysis. The

authors analyzed 57 natural *S. pombe* isolates and found MLP formation to widely vary across different nutrient and drug conditions. The authors demonstrate that MLP formation correlated with expression levels of the transcription factor gene *mbx2* and several flocculins. The authors also show that *Cdk8* kinase module and *srub11* deletions also resulted in MLP formation. The experimental design is logic, the manuscript is well-written and organized. I have a few concerns that should be addressed before the publication.

Major points:

1) Line 61-62, how did the authors grow yeast cells in the liquid medium? Shaking or static? If shaking, the nutrient should be even distributed in the medium.

If static culture, most single yeast cells could precipitate on the bottom, how do you address the advantage of flocculation for increasing the sedimentation? In addition, under static culture, the bottom will have less air than the top medium, how to balance the air and nutrients?

Reply: In line 61-62 of our original manuscript we stated that “Similarly, flocculation could increase sedimentation in liquid media, thereby assisting the search for more nutrient-rich or less stressful environments (4)”.

Our intent was to speculate on the advantages of multicellular-like growth, and cited a review article which has mentioned sedimentation. After further consideration, we decided that this is a minor point and is rather speculative, and removed it altogether from the manuscript.

In response to the Reviewer’s question about how cells were grown in liquid medium, throughout the paper we used shaking cultures for our flocculation assays and for pre-cultures. We have made this more clear in the text where it was ambiguous (e.g. line 189, throughout the methods section, and in the legend of Fig. 2A).

2) Line 555, it will be interesting to test whether overexpression of *mbx2* could cause flocculation in YES medium. In Figure 3D, the authors use two control strains, but only one *mbx2* OE strain, *mbx2* OE should be tested in both strains. In addition, did the authors transform empty plasmid into the control strains, please indicate in the figure.

In this experiment, *mbx2* was overexpressed using a thiamine-repressible *nmt1* promoter, which is a standard construct in fission yeast studies. Assaying MLP formation was not feasible in YES with this strain, because YES is a rich media made up of yeast extract which contains thiamine. Thus, we could not remove thiamine from the media to trigger *mbx2* overexpression.

In order to test the influence of *mbx2* overexpression in YES, we constructed strains in which *mbx2* was integrated into the genome and expression was driven by the *rpl2102* promoter, which has been shown to provide constitutive moderate expression levels (12). We observed strong flocculation in both EMM and YES (Fig 3D, New Supplementary Figure 7). We did not see strong flocculation in a control in which GFP was expressed under the *rpl2102* promoter. The flocculation phenotype was so strong that our original adhesion assay protocol required modification for this experiment, including resuspension in 10 mM EDTA before re-plate (Methods). We observed strong adhesion for the *mbx2* overexpression strains (Fig 3D), but not for control strains in YES. We could not check adhesion in EMM for those strains because cells pinned on EMM did not survive resuspension in EDTA.

We performed these experiments in two backgrounds, 968 *h90* (JB50), which is one of the parental strains of the segregant library analysed in Figure 3 and 972 *h-* (JB22), which is an appropriate background for the gene deletion collection.

We have replaced the data from the original Figure 3D with the new adhesion assay and added New Supplementary Figure 7 to the manuscript (Lines 236-244).

This result also helped us to further refine our model for the pathway. We can now say that the repression of MLPs in rich media must act via Mbx2, as overexpression of *mbx2* is sufficient to abolish it, and is likely to act transcriptionally (if it acted on the protein level, the mild overexpression would likely not have led to the phenotype) (Figure 6, Lines 554-556 in the discussion)

New Supplementary Figure 7: Overexpression of *mbx2* drives MLP formation.

(A) Microscopy and (B) flask images illustrating the effects of *mbx2* overexpression on cells grown in EMM or YES for both the *h-* (JB22) and *h90* (JB50) backgrounds and the *h90* JB50 background. Flask images were from overnight cultures, and microscope images were from overnight cultures after dilution into fresh media for two hours. The flask images of *prRPL2102-mbx2* (JB22) are duplicated and also shown in Figure 3. Scale bars are 75 μ m.

Figure 3D: Bar plot of \log_2FC in gene expression during 2 days of phosphate starvation for selected genes, including the transcription factor *mbx2* and various flocculins. Data from (81). Red bars represent significant \log_2FC values.

3) Line 600-601, the authors may do the backcross of *srb11Δ::Kan* to exclude the possibility caused by other mutations.

Reply: We thank the reviewer for noticing our concern about suppressor mutations arising in the *srb11Δ* strain obtained from our deletion library. This initial concern arose following the observation that while qualitatively the *srb11Δ::Kan* and *srb11Δ*(CRISPR) strains were both strongly adhesive, there was a minor quantitative difference in their adhesion.

As we obtained this strain from an *h+* deletion library strain backcrossed with a prototrophic *h-* strain (JB22) in order to restore auxotrophies (13), the chances for a suppressor mutation to arise are very low. We have therefore removed that language from our text. We now suspect that a more likely explanation for this small difference could be the strain background, as our CRISPR engineered strain was made in a JB50 background which has the *h90* mating type, while the deletion library strains are *h-* without auxotrophic markers.

We would like to emphasize, however, that despite this quantitative difference in the adhesion phenotype between the two *srb11Δ* strains, they both have a large increase in the adhesion phenotype relative to the respective wild-type strains. To address this point, we have removed

the unnecessary statistical comparison of these two deletion strains and focused on their qualitatively high levels of adhesion in the text (lines 267-269) and in our Revised Supplementary Figure 6D.

Minor points:

1) Line 506, what are the growth conditions of cells in Figure 2A? Did the authors use the liquid or solid medium? Please mention in the Methods or figure legends.

Reply: We have updated the manuscript to include the relevant details in the text (line 189), figure caption for Fig. 2A and in the methods section (lines 829-831).

2) Line 533-535, please explain why the strains exhibiting strong adhesion have a decreased growth rate. Is there any related research? Please add some references.

Reply: Please see reply to Reviewer 1, comment 5.

****Referees cross-commenting****

I agree with most of the comments from other reviewers. This publication may indeed be of interest to a minor area. But the results and the interpretations of the data are interesting and warranted, the findings are scientifically important.

Significance

The authors did many large-scale screens and bioinformatic analyses. The experiments in the manuscript are generally logical and sound. This study is useful for deciphering the mechanism of multicellular-like phenotype formation in the fission yeast, with some implications for some other organisms.

Reviewer #3 (Evidence, reproducibility and clarity (Required)):

Summary:

Using a variety of targeted and genome wide analyses, the authors investigate the basis for "multicellular-like phenotypes" in *S. pombe*. Authors developed several methodologies to detect and quantify "multicellular-like phenotypes" (flocculation, aggregation...) and defined genes involved in these processes in laboratory and wild *S. pombe*.

SECTION A - Evidence, reproducibility and clarity

This is a very solid manuscript that is well-written and supported by convincing data. While one can imagine many additional experiments, the manuscript stands on its own and presents a quite exhaustive analysis of the area. I commend the author for their rigorous work and clear presentation. There are only a few minor points that warrant comments or corrections:

- Supplementary Figure 1 is a typical example of the "necessity" to have statistics and P-values everywhere. The data are convincing but what is the evidence that the Filtering assay and the Plate-reader assay values should be linearly related? Let's imagine that

Plate-reader assay value is proportional to the square of the Filtering assay value. What would be the Pearson R and P-value in this case? What is most appropriate? Why would one use a linear correlation? What is the "real" significance?

Reply: We thank the reviewer for pointing out that the data in Supplementary Figure 1 does not appear to be linear and, therefore, reporting the Pearson correlation coefficient may not be the best way to represent the relationship between the two assays. The nonlinear nature of this data could indicate that

- 1) The filtering assay saturates before the plate reader assay, and is less able to distinguish between strains that flocculate strongly and
- 2) The filtering assay may be more sensitive for strains that show lower levels of flocculation.

In general, we observed fewer strains with intermediate phenotypes for both assays, making it difficult to ascertain the true relationship between them; however, we believe that the key result is that the strains with the highest level of flocculation have the highest values in both assays. To capture this aspect of the data, we now report the Spearman correlation which is non-parametric and indicates how similar the ranking of each strain is based on both assays. With the alternative hypothesis being that the correlation is > 0 , we report a Spearman correlation coefficient of 0.24 and a P-value of 0.04 (lines 823-826)

- Minor points:

* They are several "personal communications" in the manuscript (page 11, page 18, page 23). It should be checked whether this is accepted in the journal that publishes this manuscript.

Reply: We thank the reviewer for highlighting this issue. We had three instances of "personal communications" in our original submission.

The first instance was an acknowledgement for advice on our DNA extraction protocol from Dan Jeffares. We now include this in the Acknowledgements section instead.

The second communication with Angad Garg described that they observed flocculation while growing cells in phosphate starvation conditions, which was not reported in their publication (14). Though we appreciate their willingness to share unpublished data with us, we have removed this observation from our manuscript and instead rely only on our own observations and arguments based on their published RNA-seq data to make our point.

The third personal communication with Olivia Hillson supplements a minor hypothesis, namely that deletion of SPNCRNA.781 might cause MLP formation by affecting the promoter of *hsr1*, for which we had access to unpublished CHIP-seq data, showing its binding to flocculins. Recently published work from a different group (15) also suggests this link between *hsr1* and flocculation and is now discussed in our manuscript instead of the result based on unpublished data obtained from personal communication at Lines 397-398.

* Page 4 check "a few regulators"

Reply: For clarity, this has now been changed to "several regulatory proteins" at Line 108. The specific proteins we are referring to are highlighted in Figure 1C.

* Page 19, line 567: "remaining 8 strains" may be confusing as Material and Methods states "remaining 10 strains".

Reply: Two of the 10 strains were found to be redundant after sequencing as explained in the Methods (Lines 930-934). Therefore, we only added 8 new strains to the analysis. We thank the reviewer for highlighting this as a potential source of misunderstanding, and clarified this point in the text (Lines 247-250 and in the methods).

****Referees cross-commenting****

I concur with most comments. Overall, the reviewers agree that this is a solid piece of work that could benefit from minor modifications and should be published. I reiterate that, for me, despite its quality, this publication will only be of interest to specialists.

Reviewer #3 (Significance (Required)):

A limited number of studies have investigated "multicellular-like phenotypes" in *S. pombe*. This manuscript brings therefore new and solid information. Yet, despite an impressive amount of work, our conceptual advance in understanding this process and its phylogenetic conservation remains limited. This is probably best illustrated in the figure 6 that summarize the study and contains 3 question marks and an additional unknown mechanism. (Most of the solid arrows in this figure correspond to interactions within the Mediator complex that were well known before this study.) In addition, while only few studies have been published in this area, the authors' findings are often only bringing additional support to already published observations.

Overall, while this manuscript will be of interest to a restricted group of aficionados, it will most likely not attract the attention of a wide readership.

Reviewer #4 (Evidence, reproducibility and clarity (Required)):

In this manuscript, the authors explore how multicellular-like phenotypes (MLPs) arise in the fission yeast *S. pombe*. Although yeasts are characterized as unicellular fungi, diverse species show MLPs, including filamentous growth on agar plates and flocculation in liquid media. MLPs may provide certain advantages in nutritionally poor conditions and protection against external challenges, upon which natural selection can then act. Previous work on MLPs has mostly been carried out in the budding yeasts *S. cerevisiae* and *C. albicans*, and little was known about these behaviors in *S. pombe*. The authors thus set out to investigate both genetic and environmental regulators of MLP formation.

First, their analysis of published data revealed a limited number of shared regulators of MLP between *S. pombe*, *S. cerevisiae*, and *C. albicans*, although the cell adhesion proteins themselves are largely not conserved. Next, the authors screened a set of non-clonal natural isolates using two high-throughput assays that they developed and found that MLPs vary in strains and depending on nutrient conditions. Focusing on a natural isolate that showed both adhesion on agar plates and flocculation in liquid medium, they then analyzed a segregant library generated from this and a laboratory strain using their assays. Using QTL analysis, they uncovered a frameshift in the *srb11* gene, which encodes a subunit of the Mediator complex, as the likely causal inducer of MLP. This was confirmed by additional analyses of

strains lacking *srb11* or other members of Mediator. Furthermore, the authors showed that loss of *srb11* function resulted in the upregulation of the Mbx2 transcription factor, which was both necessary and sufficient for MLP formation in this background. Finally, screening of two additional yeast strain collections (gene and long intergenic non-coding RNA deletion) identified both known and novel regulators representing different pathways that may be involved in MLP formation.

Altogether, this study provides new perspectives into our understanding of the diverse inputs that regulate multicellular-like phenotypes in yeast.

Major comments:

- The methods for screening for adhesion and flocculation are well described, with representative figures that show plates and flasks. However, there are few microscopy images of cells, and it would be interesting and helpful for the reader to have an idea of how cells look when they exhibit MLPs. For instance, are there any differences in cell shape or size when strains present different degrees of adhesion or flocculation? In addition, the authors mention that mutants with strong adhesion generally had lower colony density and are likely to be slower growing. Although their analyses suggest otherwise (page 22), this has a potential for introducing error in their observations, and including images of the adhesion/flocculation phenotypes may provide further support for their conclusions. I suggest that the authors present microscopy images 1) similar to what is shown for JB759 in Figure 2A and 2) of cells growing on agar in the adhesion assay. This could be included for the different Mediator subunit deletions that they tested, where there appear to be varying phenotypes. It could also be informative for a subset of the 31 high-confidence candidates that they identified in their screen.

Reply: We thank the reviewer for highlighting the need for further microscopic characterisation of MLP forming strains. We therefore now include images of JB914, JB953 (New Supplementary Figures 4, Figure 2E) in liquid media in EMM, EMM-Phosphate, and YES; an *srb11* deletion strain (Figure 3F), and *mbx2* overexpression strains (New Supplementary Figure 7).

- Upon identifying a frameshift in *srb11* that is responsible for the MLP, the authors assessed whether deletion of other Mediator subunits would result in the same phenotype. They found that *srb10* and *srb11* deletions both flocculate and show adhesion, while other mutants had milder phenotypes. However, the authors also found that a new deletion of *srb11* that they generated had a stronger adhesion phenotype than the *srb11* deletion from the prototrophic deletion library, which was attributed this the accumulation of suppressor mutations in the strains of the deletion collection. As the authors make clear distinctions between the phenotypes of different Mediator mutants, I suggest generating and analyzing "clean" deletions of the 6 other subunits that they tested. This would strengthen their conclusion and help to rule out accumulated suppressors as the cause of the differences in the observed phenotypes.

Reply: We thank the reviewer for noticing our concern about suppressor mutations in the manuscript. As we describe above in response to a similar question from reviewer 2, as the prototrophic deletion library from which we extracted the Mediator deletion strains had been backcrossed during its construction (13), we no longer suspect that small difference between the *srb11Δ::Kan* strain from the deletion library and the newly created *srb11Δ* (CRISPR) strains is due to suppressor mutations. Rather, we think they may be a result of the difference in genetic

background and possibly mating type between the two strains. We also want to emphasize that this difference is small compared to the difference between the adhesion ratios of the *srb11Δ* strains and their respective control strains.

Nevertheless, we made clean, independent Mediator mutants for 5 out of 6 Mediator genes tested (*med10Δ*, *med13Δ*, *med19Δ*, *med27Δ*, and *srb10Δ*) as well as an additional mutant that we didn't have in our library, *med12Δ* (Figure R9). When running the assay on these new strains we got an overall lower dynamic range, possibly due to variations in the water flow rate relative to the first assay. However, we saw a strong phenotype for both library and our own *srb10Δ* and CRISPR *srb11Δ* strains. We did not see a significant increase in adhesion for the other Mediator deletion mutants in EMM relative to wild type with the exception of for *med10Δ* in both the library strain and for our clean mutant, for which we did not observe a phenotype in our previous experiment. We included the experiment for the newly created mutants as New Supplementary Figure S6E and described them in lines 276-281 in our revised manuscript.

New Supplementary Figure S6E: Box plot showing adhesion values from validation cohort of fresh Mediator gene deletion strains on EMM and YES as indicated. Each dot represents a replicate.

Minor comments:

- One point that recurs in the manuscript is the idea that mutations that give rise to strong MLPs also generally lead to slower growth, representing a potential trade-off. This idea could be reinforced with measurements of growth rate or generation time by optical density or cell number, for instance, rather than comparisons of colony density. Also, it would be interesting to mention if the slow growth phenotype is only observed in MLP-inducing conditions or also in rich medium.

Reply: As described above in response to item 5 from Reviewer 1, we have conducted growth assays in liquid media for *srb10Δ*, *srb11Δ*, and other mutants from our adhesion screen (*tlg2Δ*, *rpa12Δ*, *mus7Δ* and *kgd2Δ*) that showed a similar phenotype to those genes in both minimal (EMM) and rich (YES) media. We observe that in rich media, *srb10Δ* and *srb11Δ* cells grow similarly to control strains, and they exhibit a lower decrease in growth rate than the other similarly adhesive strains. Both *mus7Δ* and *kgd2Δ* cells grow more slowly, even in rich media.

We have also added data on the tradeoff between growth and adhesion based on growth on solid media from (11) for all mutants identified in our screen (New Supp Fig 12B)).

Thus, the relationship between slow growth and clumpiness depends on the mutation, and specifically, mutations of the Mediator, including those to *srb11* and *srb10*, seem to decrease the impact of any tradeoff between growth and adhesion.

- The authors show that the MLPs of the *srb10* and *srb11* deletions occur through *mbx2* upregulation. Do the varying strengths of the phenotypes of the strains lacking different Mediator subunits correlate with *mbx2* levels in these backgrounds?

Reply: There is some evidence from previous work that the relationship between the strength of the MLPs and the expression of *mbx2* may not be perfectly proportional. In (16), *med12Δ* had a higher (though qualitatively comparable) level of *mbx2* upregulation than *srb10Δ* (New Supp Fig 8E), even though that paper reported a milder phenotype for *med12Δ* than for *srb10Δ* cells. We did not observe a significant increase in adhesion in our *med12Δ* strain (New Supp Fig 6D). This suggests that in the case of these mutants, it is not simply the level of *mbx2* that controls MLP formation, but that there are likely additional regulatory mechanisms. We have added some discussion on this context in the manuscript (lines 545-547).

****Referees cross-commenting****

I agree overall with the comments and suggestions from the other reviewers. The revision would require only minor modifications. The paper is interesting both for the combination of methodologies used and its findings, and I believe that it would benefit a growing community of researchers.

Reviewer #4 (Significance (Required)):

This study employed a variety of methods that allowed the authors to uncover previously unknown regulators of MLPs. Taking advantage of the diversity of natural fission yeast isolates as well as the constructed gene and non-coding RNA deletion collections, the authors identified novel genetic determinants that give rise to MLPs, opening new avenues into this exciting area of research. The overall conclusions of the work are solid and supported by the reported results and analyses. This study will be appreciated by a broad audience of readers who are interested in understanding how organisms respond to environmental challenges as well as how MLPs may result in emergent properties that play key roles in these responses. Some of the limitations of the work are described above, with recommendations for addressing these points.

Keywords for my field of expertise: fission yeast, cell cycle, transcription, replication.

1. Brysch-Herzberg M, Jia GS, Seidel M, Assali I, Du LL. Insights into the ecology of *Schizosaccharomyces* species in natural and artificial habitats. *Antonie Van Leeuwenhoek*. 2022 May 1;115(5):661–95.

2. Jeffares DC, Rallis C, Rieux A, Speed D, Převorovský M, Mourier T, et al. The genomic and phenotypic diversity of *Schizosaccharomyces pombe*. *Nat Genet.* 2015 Mar;47(3):235–41.
3. Ratcliff WC, Denison RF, Borrello M, Travisano M. Experimental evolution of multicellularity. *Proc Natl Acad Sci.* 2012 Jan 31;109(5):1595–600.
4. Smukalla S, Caldara M, Pochet N, Beauvais A, Guadagnini S, Yan C, et al. FLO1 is a variable green beard gene that drives biofilm-like cooperation in budding yeast. *Cell.* 2008 Nov 14;135(4):726–37.
5. Lorenz MC, Heitman J. Yeast pseudohyphal growth is regulated by GPA2, a G protein alpha homolog. *EMBO J.* 1997 Dec 1;16(23):7008–18.
6. Ignacia DGL, Bennis NX, Wheeler C, Tu LCL, Keijzer J, Cardoso CC, et al. Functional analysis of *Saccharomyces cerevisiae* FLO genes through optogenetic control. *FEMS Yeast Res.* 2025 Sept 24;25:foaf057.
7. Wang Z, Xu W, Gao Y, Zha M, Zhang D, Peng X, et al. Engineering *Saccharomyces cerevisiae* for improved biofilm formation and ethanol production in continuous fermentation. *Biotechnol Biofuels Bioprod.* 2023 July 31;16(1):119.
8. Koschwanez JH, Foster KR, Murray AW. Improved use of a public good selects for the evolution of undifferentiated multicellularity. *eLife.* 2013 Apr 2;2:e00367.
9. Westman JO, Mapelli V, Taherzadeh MJ, Franzén CJ. Flocculation Causes Inhibitor Tolerance in *Saccharomyces cerevisiae* for Second-Generation Bioethanol Production. *Appl Environ Microbiol.* 2014 Nov;80(22):6908–18.
10. Li R, Li X, Sun L, Chen F, Liu Z, Gu Y, et al. Reduction of Ribosome Level Triggers Flocculation of Fission Yeast Cells. *Eukaryot Cell.* 2013 Mar;12(3):450–9.
11. Rodríguez-López M, Bordin N, Lees J, Scholes H, Hassan S, Saintain Q, et al. Broad functional profiling of fission yeast proteins using phenomics and machine learning. Marston AL, James DE, editors. *eLife.* 2023 Oct 3;12:RP88229.
12. Hebra T, Smrčková H, Elkatmis B, Převorovský M, Pluskal T. POMBOX: A Fission Yeast Cloning Toolkit for Molecular and Synthetic Biology. *ACS Synth Biol.* 2024 Feb 16;13(2):558–67.
13. Malecki M, Bähler J. Identifying genes required for respiratory growth of fission yeast. *Wellcome Open Res.* 2016 Nov 15;1:12.
14. Garg A, Sanchez AM, Miele M, Schwer B, Shuman S. Cellular responses to long-term phosphate starvation of fission yeast: Maf1 determines fate choice between quiescence and death associated with aberrant tRNA biogenesis. *Nucleic Acids Res.* 2023 Feb 16;51(7):3094–115.
15. Ohsawa S, Schwaiger M, Iesmantavicius V, Hashimoto R, Moriyama H, Matoba H, et al. Nitrogen signaling factor triggers a respiration-like gene expression program in fission yeast. *EMBO J.* 2024 Oct 15;43(20):4604–24.
16. Linder T, Rasmussen NN, Samuelsen CO, Chatzidaki E, Baraznenok V, Beve J, et al. Two conserved modules of *Schizosaccharomyces pombe* Mediator regulate distinct cellular pathways. *Nucleic Acids Res.* 2008 May;36(8):2489–504.

February 19, 2026
RE: GENETICS-2026-309023

Dr. Benjamin Heineike

Dear Dr. Heineike:

Congratulations, your manuscript titled "Genetic and environmental determinants of multicellular-like phenotypes in fission yeast" is accepted for publication in GENETICS! Many thanks for submitting your research to the journal.

I find that you have done an excellent job of addressing the comments from 4 reviewers at Review Commons and share their enthusiasm for publishing the work. Thank you for submitting this work to GENETICS.

To Proceed to Publication:

1. Format your article according to GENETICS style: <https://academic.oup.com/genetics/pages/author-guidelines>
2. Ensure that you comply with data and community resource citation guidelines: <https://academic.oup.com/genetics/pages/author-guidelines#section-5-9-2>
3. Upload your final files at <https://genetics.msubmit.net>
4. Add oupsupport@scipris.com and genetics.oup@novatechset.com (or the domains @scipris.com and @novatechset.com) to your email program's "safe senders" list. You will be contacted by both at various points during the production process.

Notes:

- Your currently-accepted manuscript (unedited, as submitted, reviewed, and accepted) will be published at GENETICS and deposited into PubMed as an Advance Access article. Notify sourcefiles@thegsajournals.org before signing your license if you do not wish to publish your article via Advance Access.
- We invite you to submit an original color figure related to your paper for consideration as cover art. Please email your submission to the editorial office or upload it with your final files. You can submit a small-sized image for evaluation, and if selected, the final image must be a TIFF file 2513px wide by 3263px high (8.375 by 10.875 inches; resolution of 600ppi). Please avoid graphs and small type.
- After files are sent to Oxford University Press we use SciPris to manage article licensing and payment. If you do not have a SciPris account, you will receive an email from no-reply@scipris.com to sign up to use Oxford University Press' author portal. After logging in, follow the online instructions to sign your license and arrange any payment due.

If you have any questions or encounter any problems while uploading your accepted manuscript files, please email the editorial office at sourcefiles@thegsajournals.org.

Sincerely,

Patricia Wittkopp
Associate Editor
GENETICS

Approved by:
Anthony Long
Senior Editor
GENETICS